METHODS AND RESOURCES

# High-throughput characterization of photocrosslinker-bearing ion channel variants to map residues critical for function and pharmacology

Nina Braun[ID][1], Søren Friis[2], Christian Ihling[3], Andrea Sinz[ID][3], Jacob Andersen[1¤], Stephan A. Pless[ID][1]*

1 Department of Drug Design and Pharmacology, University of Copenhagen, Copenhagen, Denmark,
2 Nanion Technologies GmbH, Munich, Germany, 3 Department of Pharmaceutical Chemistry & Bioanalytics, Center for Structural Mass Spectrometry, Institute of Pharmacy, Martin Luther University Halle-Wittenberg, Halle (Saale), Germany

¤ Current address: Lundbeck Research, H. Lundbeck A/S, Valby, Denmark
* stephan.pless@sund.ku.dk

**Data Availability Statement:** All raw data, including original Western blots, has been deposited to an online repository: https://zenodo.

## Abstract

Incorporation of noncanonical amino acids (ncAAs) can endow proteins with novel functionalities, such as crosslinking or fluorescence. In ion channels, the function of these variants can be studied with great precision using standard electrophysiology, but this approach is typically labor intensive and low throughput. Here, we establish a high-throughput protocol to conduct functional and pharmacological investigations of ncAA-containing human acid-sensing ion channel 1a (hASIC1a) variants in transiently transfected mammalian cells. We introduce 3 different photocrosslinking ncAAs into 103 positions and assess the function of the resulting 309 variants with automated patch clamp (APC). We demonstrate that the approach is efficient and versatile, as it is amenable to assessing even complex pharmacological modulation by peptides. The data show that the acidic pocket is a major determinant for current decay, and live-cell crosslinking provides insight into the hASIC1a–psalmotoxin 1 (PcTx1) interaction. Further, we provide evidence that the protocol can be applied to other ion channels, such as P2X2 and GluA2 receptors. We therefore anticipate the approach to enable future APC-based studies of ncAA-containing ion channels in mammalian cells.

## Introduction

Genetic code expansion approaches allow the incorporation of noncanonical amino acids (ncAAs) with unique chemical properties into proteins. Over the past 2 decades, this method has greatly facilitated protein modification and functionalization beyond the confines of the genetic code [1]. Ion channels have proven highly suited to ncAA incorporation, as evidenced by the success in introducing photocrosslinking, photoswitchable, or fluorescent ncAAs into numerous members of this large and diverse protein family [2–4]. Among the ncAA subclasses, photocrosslinkers have proven particularly versatile, as they allow for the trapping of

org/record/4906985. All analysed data are within the paper and its Supporting Information files.

**Funding:** We acknowledge the Lundbeck Foundation (R139-2012-12390 to SAP and R218-2016-1490 to NB), the Boehringer Ingelheim Fond (to NB) and the Deutsche Forschungsgemeinschaft (#391498659 to AS) for financial support. The funders had no role in study design, data collection and analysis, decision to publish, or preparation of the manuscript.

**Competing interests:** I have read the journal's policy and the authors of this manuscript have the following competing interests: Søren Friis is a full-time employee of Nanion Technologies. The other authors declare no competing interests.

**Abbreviations:** aaRS, aminoacyl-tRNA synthetase; AMPAR, α-amino-3-hydroxy-5-methyl-4-isoxazolepropionic acid receptor; APC, automated patch clamp; ASIC, acid-sensing ion channel; AzF, 4-Azido-L-phenylalanine; BigDyn, big dynorphin; Bpa, 4-Benzoyl-L-phenylalanine; BSA, bovine serum albumin; cDNA, complementary DNA; DN-eRF, dominant negative eukaryotic release factor; ECD, extracellular domain; FACS, fluorescence-activated cell sorting; hASIC1a, human acid-sensing ion channel 1a; hSERT, human serotonin transporter; ICD, intracellular domain; IRES, internal ribosome entry site; ncAA, noncanonical amino acid; PcTx1, psalmotoxin 1; SD, standard deviation; Se-AbK, (R)-2-Amino-3-{2-[2-(3-methyl-3H-diazirin-3-yl)-ethoxycarbonylamino]-ethylselanyl}-propionic acid; SSD, steady-state desensitization; TEVC, two-electrode voltage clamp; TMD, transmembrane domain; WT, wild type.

ion channels in certain conformational states [5–8], capturing of protein–protein interactions [9–12], and covalent linking of receptor–ligand complexes to delineate ligand binding sites [13–17].

Typically, incorporation of ncAAs is achieved by repurposing a stop codon to encode for an ncAA supplied by an orthogonal tRNA/aminoacyl-tRNA synthetase (aaRS) pair. But the incorporation efficiency can be variable, and unspecific incorporation of naturally occurring amino acids can result in inhomogeneous protein populations [2]. Verification of site-specific ncAA incorporation can therefore be laborious and time consuming, especially in combination with detailed functional characterization. As a result, most studies have focused on only a limited number of incorporation sites, and the evaluation of potential functional or pharmacological effects of ncAA incorporation often remained minimal. In principle, automated patch clamp (APC) devices offer fast and efficient high-throughput testing and have recently gained increasing popularity for electrophysiological interrogation of a diverse set of ion channels [18–22]. However, a combination of low efficiency of transient transfection in mammalian cells and limited ncAA incorporation rates have thus far prevented functional screening of ncAA-containing ion channel variants on APC platforms.

Here, we sought to overcome these limitations by developing a fluorescence-activated cell sorting (FACS)-based approach to enrich the population of transiently transfected cells expressing ncAA-containing ion channels. Using the human acid-sensing ion channel 1a (hASIC1a) as an example, we incorporated 3 different ncAA photocrosslinkers (4-Azido-L-phenylalanine (AzF), 4-Benzoyl-L-phenylalanine (Bpa), and (R)-2-Amino-3-{2-[2-(3-methyl-3H-diazirin-3-yl)-ethoxycarbonylamino]-ethylselanyl}-propionic acid (Se-AbK)) at 103 positions throughout its intracellular domain (ICD), extracellular domain (ECD), and transmembrane domain (TMD).

Acid-sensing ion channels (ASICs) are trimeric ligand-gated ion channels that open a weakly sodium selective pore in response to proton binding to the so-called acidic pocket and likely other sites in the ECD [23]. Apart from contributions to synaptic plasticity [24,25], ASICs have recently gained increasing attention as potential drug targets for pain and stroke [26–35]. The 6 different human ASIC isoforms (ASIC1a, ASIC1b, ASIC2a, ASIC2b, ASIC3, and ASIC4) are modulated by an impressive array of neuropeptides and venom-derived toxins that bind to the large ECD [24,36,37]. Intriguingly, the extent and type of modulation (e.g., inhibition versus potentiation) are often highly dependent on ambient proton concentration, as well as subtype and species origin [38,39]. This poses challenges for pharmacological profiling and motivates a detailed understanding of the mechanism and site of action of these peptides, to eventually generate lead compounds that could potentially target pain or stroke.

In this study, we establish a protocol to functionally screen ncAA-containing ion channels in transiently transfected cells on an APC platform. The 384-well setup of the SyncroPatch 384PE (Nanion Technologies, Germany) allows the efficient characterization of 309 hASIC1a variants, and we show that ncAA incorporation is tolerated in over 50% of the positions. Incorporation of bulky ncAA photocrosslinkers generally results in lower pH sensitivity, especially around the acidic pocket, where ncAA incorporation also greatly accelerates current decay kinetics. We further demonstrate differential channel modulation by the neuropeptide big dynorphin (BigDyn; [40]) and by psalmotoxin 1 (PcTx1; [41]), a toxin derived from tarantula venom. Lastly, we turn to UV-induced photocrosslinking to covalently trap channel–toxin complexes and thus map the hASIC1a–PcTx1 interaction in live cells. Overall, our work highlights that ncAA-containing ion channels, including ASICs, ionotropic glutamate, and P2X receptors, are amenable to APC-based high-throughput screening. We further demonstrate how this approach, when used with ncAA photocrosslinkers, can be harnessed to investigate protein–peptide or protein–protein interactions in cellulo.

## Results

### Development of an APC screen to validate ncAA incorporation into hASIC1a

In order to efficiently assess functional incorporation of ncAAs into hASIC1a, we developed an APC screen to record proton-gated channel activation (Fig 1). To this end, we co-transfected 103 different hASIC1a variants containing individual TAG stop codons throughout the protein together with the suppressor tRNA/ncAA-RS pair for either AzF, Bpa, or Se-AbK and a GFP reporter carrying a TAG at Y40 (for Bpa and Se-AbK) or Y151 (for AzF, as we observed a higher degree of unspecific incorporation in the Y40TAG variant with AzF) into ASIC1a-KO HEK 293T cells [17,42–44]. The corresponding ncAA was supplied in the cell culture medium 6 hours after transfection or omitted from the experiment in incorporation control samples. To increase cell viability and uptake efficiency, we synthesized the methylester derivates of AzF and Bpa [8,45]. This allowed us to supplement the cell media with 50- and 100-fold lower ncAA concentration compared to previous studies, respectively [7,13].

After 48 hours, cells grown in the presence of ncAA were sorted for green fluorescence to enrich the population of transfected cells, which were then submitted to APC to record proton-gated currents. Using GFP fluorescence as a proxy, we determined a transfection efficiency of 62.9 ± 9.5% for hASIC1a wild type (WT) and an average of 11.2 ± 5% for the ncAA variants (S1 Fig, S1 Table). Without the FACS step, the latter rate would translate into less than 10% of the APC wells being occupied by transfected cells, precluding efficient APC experiments. By contrast, the cell sorting improves occupation to around 46% of wells with successful patch also displaying proton-gated currents (62% for AzF, 29% for Bpa, and 48% for Se-AbK) and is therefore an indispensable element for the use of transiently transfected cells in APC (S1 Fig).

The 384-well system of the SyncroPatch 384PE allows for parallel concentration response curve measurements on 24 different samples, enabling us to test 11 different channel variants with corresponding incorporation controls (cells grown in the absence of ncAA), as well as

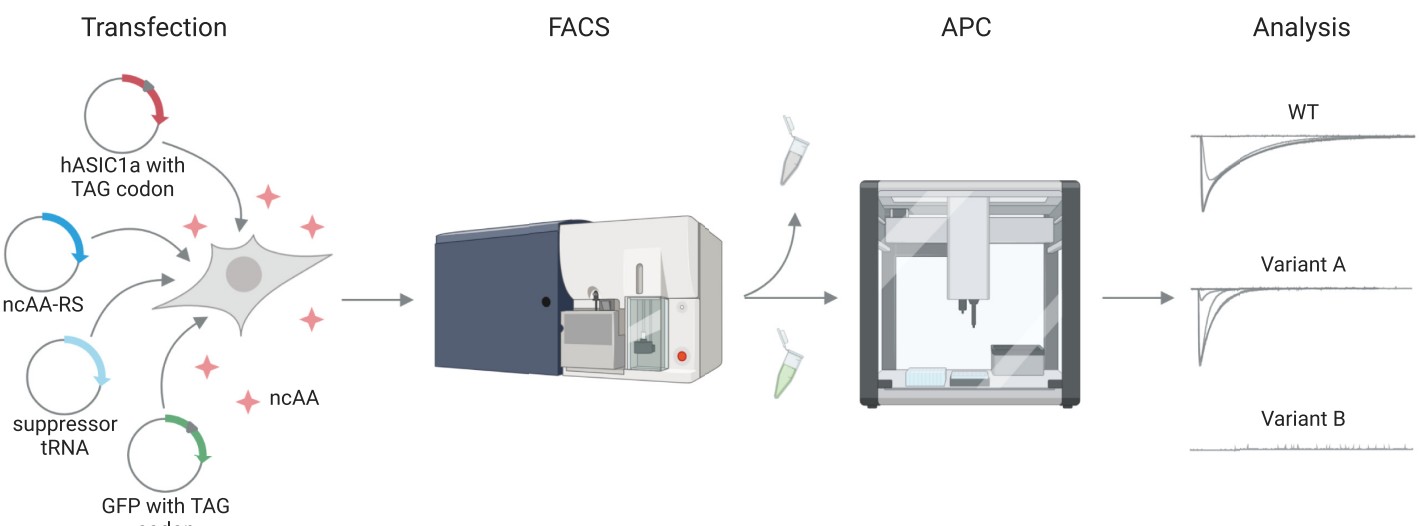

**Fig 1. Schematic illustration of the workflow to assess ncAA incorporation into hASIC1a.** HEK 293T ASIC1a-KO cells are transfected with hASIC1a containing a TAG stop codon at the site of interest, a coevolved suppressor tRNA/ncAA-RS pair, and a TAG-containing GFP reporter. ncAA is supplied in the cell culture medium. Moreover, 48 hours after transfection, cells are sorted for green fluorescence on a FACS BD Aria I, and those showing fluorescence are subjected to APC on a SyncroPatch 384PE to measure proton concentration response curves. APC, automated patch clamp; FACS, fluorescence-activated cell sorting; hASIC1a, human acid-sensing ion channel 1a; ncAA, noncanonical amino acid; WT, wild type.

hASIC1a WT and untransfected cells in less than 1 hour, with up to 16 replicates per sample. Specifically, we embarked to functionally interrogate 103 positions throughout the hASIC1a sequence: 38 positions in the N-terminal domain (S2 Fig), 24 positions in the TMD and interface region (S3 Fig), 29 positions in the carboxyl-terminal domain (S4 Fig), and 12 positions around the acidic pocket (Fig 3, S8 Fig). The current traces in Fig 2A show typical pH-induced inward currents of hASIC1a WT with a $pH_{50}$ of 6.64 ± 0.12 ($n$ = 182), in line with previous studies [46,47], as well as a variant with lower proton sensitivity containing AzF in the acidic pocket (T236AzF, $pH_{50}$ 6.17 ± 0.14, $n$ = 10). Interestingly, the incorporation of Bpa, AzF, and Se-AbK at position W46 did not result in proton-gated currents (Fig 2A, S3 Fig), despite a previous report showing functional incorporation of a bulky ncAA at this conserved Trp in the M1 helix [48]. We analyzed all variants for mean peak current size and $pH_{50}$ to compare incorporation efficiency and proton sensitivity, respectively (S2–S4 and S8 Figs, S1 Table). Furthermore, we routinely assessed the extent of tachyphylaxis [49], and variants displaying >20% current decrease after reaching the peak current are indicated in Fig 3 and S2–S8 Figs as well as S1 Table.

To provide a comprehensive overview, we mapped incorporation patterns for the 3 photocrosslinkers onto snake plots schematically depicting an ASIC1a subunit (Fig 2B–2D). We defined specific incorporation (circles with dark color shade) as proton-gated currents of >1 nA observed in the presence of ncAA and minimal (<500 pA) proton-gated currents in the absence of ncAA. If currents >500 pA were observed in the absence of ncAA, incorporation was considered unspecific (circles with lighter color shade), while positions labeled in gray did not yield substantial currents in either condition (<1 nA). However, we cannot exclude the possibility of underestimating the degree of unspecific incorporation, as enriching transfected cells grown in the absence of ncAA by FACS was not feasible due to the low number of cells displaying GFP fluorescence (2.2 ± 1.7%). On the other hand, by defining incorporation as not successful for currents <1 nA, we are aware that we may have potentially excluded variants in which specific ncAA incorporation resulted in reduced open probability or lower conductance.

As is apparent from the snake plots, we observed robust incorporation in the N-terminus, around the acidic pocket, and in the proximal carboxyl terminus. Indeed, among the 80 positions tested up to and including L465, AzF resulted in functional channel variants in 61% of cases, compared to 50% for Bpa and 44% for Se-AbK (Fig 2E).

By contrast, all 3 crosslinkers showed mostly unspecific incorporation distal of L465, with WT-like current phenotypes from position 467 onward (S4 and S5A–S5C Figs). This led us to hypothesize that channel constructs truncated in this region are functional. To investigate this further, we inserted an additional TGA stop codon for several variants, confirmed channel truncation by comparing molecular weight on a western blot and measured concentration response curves in APC and two-electrode voltage clamp (TEVC) (S5D and S5E Fig). We found that channels truncated after H463 or K464 yielded no current in either APC or TEVC, but truncation after L465 produced a variant with strong tachyphylaxis in HEK 293T cells (S5D Fig), and truncation after C466 or R467 resulted in channels with WT-like proton sensitivity in both APC and TEVC. We conclude that the carboxyl terminus distal of position 465 is not essential for proton-gated channel activity and that it is not possible to differentiate between currents originating from truncated and full-length protein to evaluate ncAA incorporation. We therefore added a carboxyl-terminal 1D4-tag to the hASIC1a construct to selectively purify full-length protein and compare the amounts in cells grown in the presence or absence of ncAA. This strategy confirms efficient incorporation in the distal carboxyl terminus (S6A Fig). Additionally, liquid chromatography/tandem mass spectrometry data revealed that Bpa can be specifically incorporated at positions distal of L465 (A480, S6B Fig).

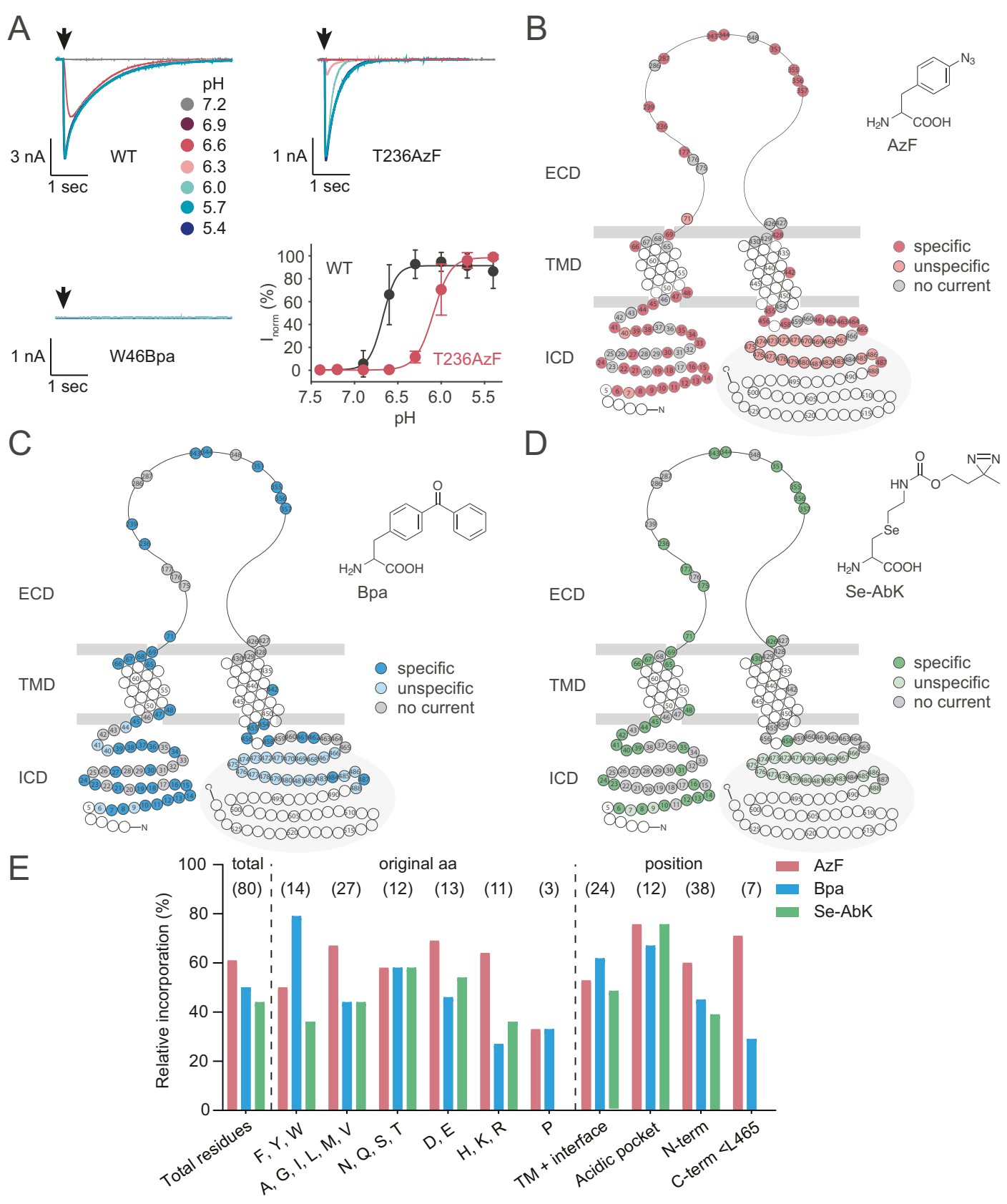

**Fig 2. Incorporation of ncAA crosslinkers is tolerated in all domains of hASIC1a and produces functional channel variants. (A)** Representative current traces for pH response curves of hASIC1a WT, T236AzF, and W46Bpa recorded on the SyncroPatch 384PE, pH response curve in bottom right corner (WT pH$_{50}$ 6.64 ± 0.12, $n$ = 182; T236AzF pH$_{50}$ 6.17 ± 0.14, $n$ = 10). **(B–D)** Snake plot representations indicating specific, unspecific, and unsuccessful incorporation (no current) for AzF (B), Bpa (C), and Se-AbK (D). Specific incorporation (circles with darker shade) is defined as pH-dependent peak currents >1 nA observed in cells grown in the presence, but not in the absence of ncAA, whereas unspecific incorporation (circles with lighter shade) indicates that currents were observed both in the presence and absence of ncAA. Positions indicated by gray circles did not yield functional channel variants when replaced by an ncAA (no current), while those colored in white were not tested. The gray area highlights positions distal of L465. **(E)** Relative incorporation rates of AzF (red), Bpa (blue), and Se-AbK (green) at 80 different hASIC1a positions. Exchanged amino acids are grouped for original side chain properties and position within the channel, respectively. Relative incorporation rates were calculated by dividing the number of positions successfully replaced with an ncAA by the total number of positions at which incorporation was attempted. Positions distal of L465 were excluded from the analysis (highlighted in gray in B–D), as more distal deletions result in truncated, but functional channels (see S4–S6 Figs). The underlying data have been deposited at zenodo.org (https://doi.org/10.5281/zenodo.4906985; files 02–08 and 15–28). AzF, 4-Azido-l-phenylalanine; Bpa, 4-Benzoyl-l-phenylalanine; C-term <L465, carboxyl terminus up to and including L465; ECD, extracellular domain; hASIC1a, human acid-sensing ion channel 1a; ICD, intracellular domain; ncAA, noncanonical amino acid; N-term, N-terminus; Se-AbK, (R)-2-Amino-3-{2-[2-(3-methyl-3H-diazirin-3-yl)-ethoxycarbonylamino]-ethylselanyl}-propionic acid; TM, transmembrane; TMD, transmembrane domain; WT, wild type.

For the 80 positions up to and including L465, we evaluated incorporation efficiency by comparing how many positions could be functionally replaced by each of the ncAA photocrosslinkers, based on the nature of the side chain occupying the position in the native channel and the position within the protein overall. We did not find evidence for pronounced global trends, but, for instance, Bpa incorporation was tolerated best at originally aromatic side chains (79%), while replacement of basic residues was least successful (27%) (Fig 2E). The 3 tested prolines could not be exchanged for any of the ncAAs. Interestingly, and in contrast to our expectations, Se-AbK incorporation only produced functional variants in 33% of cases when replacing structurally similar Lys and Arg side chains, while success rates were higher at polar and acidic side chains (58% and 54%, respectively). AzF incorporation rates were similar throughout all protein domains, whereas Bpa was better tolerated in the transmembrane regions and less in the N-terminal and carboxyl terminal and Se-AbK incorporation in the M2 helix and carboxyl terminus was negligible (Fig 2E). Overall, incorporating the 3 photocrosslinkers produced functional variants in all protein domains, albeit with varying success rates.

Together, we show that combining FACS with APC affords the time-efficient functional characterization of over 300 hASIC1a variants and provides a versatile platform to assess successful ncAA incorporation throughout all protein domains.

To evaluate if the established APC screen can also serve as a platform for other ion channels, we applied it to selected TAG variants of the rat P2X2 and rat GluA2 receptors. Specifically, we compared currents upon exposure to 2 different concentrations of ATP or glutamate, respectively (S7 Fig, S2 Table [5,7,50]). Incorporation of AzF into position K296 of the rP2X2 receptor is unspecific, whereas that of Bpa is efficient and specific (S7A Fig). For GluA2, incorporation patterns at Y533 and S729 are identical to those observed in previous studies using manual patch clamp (S7B Fig [5,7]). Incorporation of AzF at Y533 is tolerated with currents of 1.21 ± 0.96 nA ($n$ = 30, compared to 600 ± 100 pA ($n$ = 15) reported by Poulsen and colleagues), while incorporation of Bpa does not produce functional channels. At position S729, we observe small currents of 390 ± 330 pA for AzF ($n$ = 17) and 280 ± 240 pA for Bpa ($n$ = 16, compared to 470 ± 50 pA reported by Klippenstein and colleagues for S729Bpa). Importantly, as GluA2 gating is fast compared to the perfusion speed of the SyncroPatch 384PE and Klippenstein and colleagues report increased desensitization rates for S729 variants, we preincubated cells with 100 μM cyclothiazide to slow desensitization and increase the likelihood of resolving the GluA2 peak current [51]. While our GluA2 and P2X2 data show that target-specific optimization of the ligand application protocols is required, they illustrate that our APC screening approach can be applied to a variety of different ion channels and yields results comparable to those obtained with conventional ncAA incorporation protocols.

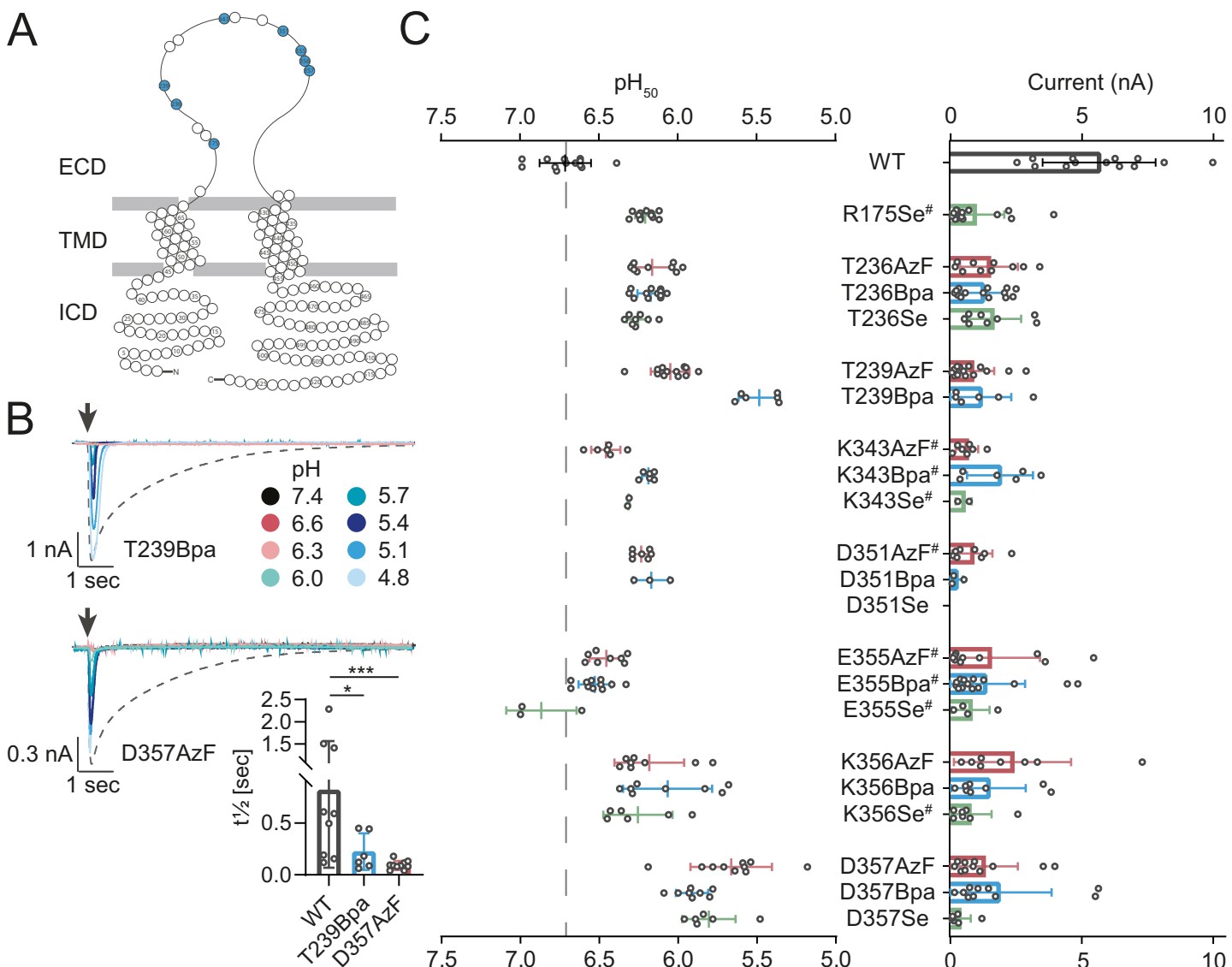

**Fig 3. Incorporation of ncAA photocrosslinkers into the acidic pocket results in channel variants with lowered proton sensitivity and accelerated current decay.**
**(A)** Snake plot of hASIC1a with the assessed positions highlighted in blue. **(B)** Representative current traces of variants T239Bpa and D357AzF as recorded on the SyncroPatch 384PE, with arrows indicating the time of the proton application. Dashed lines indicate WT current in response to pH 6.0 application. Bar graph shows mean t½ ± SD of current decay. **(C)** Incorporation of AzF (red), Bpa (blue), and Se-AbK (green) at 8 positions around the acidic pocket results in lowered proton sensitivity for several variants. Dot plots comparing $pH_{50}$ (left) and peak current sizes (right); bars indicate mean ± SD, (#) indicates >20% tachyphylaxis (see also S8 Fig and S1 Table), and (*) denotes significant difference between t½ of current decay, $p < 0.05$; (***): $p < 0.001$; Mann–Whitney test (see also S3 Table). The underlying data have been deposited at zenodo.org (https://doi.org/10.5281/zenodo.4906985; files 09, 10, and 33). AzF, 4-Azido-l-phenylalanine; Bpa, 4-Benzoyl-l-phenylalanine; ECD, extracellular domain; hASIC1a, human acid-sensing ion channel 1a; ICD, intracellular domain; ncAA, noncanonical amino acid; SD, standard deviation; Se-AbK, (R)-2-Amino-3-{2-[2-(3-methyl-3H-diazirin-3-yl)-ethoxycarbonylamino]-ethylselanyl}-propionic acid; TMD, transmembrane domain; WT, wild type.

## Photocrosslinker incorporation in the acidic pocket decreases proton sensitivity and accelerates current decay

During the design of the construct library for the APC screen, we consulted the 2.8 Å resolution structure of PcTx1 bound to chicken ASIC1 (PDB 4FZ0) to select 12 positions around the acidic pocket that are in sufficiently close proximity to potentially form covalent crosslinks with PcTx1 if replaced by an ncAA [52] (S8A Fig). Most of the resulting ncAA channel variants

were functional, but, in several instances, the initially applied proton concentration range of up to pH 5.4 did not yield saturating currents (S8B and S8C Fig). Consequently, we reevaluated these variants using a lower pH range to resolve the $pH_{50}$ and reassess peak current size (Fig 3). This allowed us to determine $EC_{50}$ values for all variants and confirmed that hASIC1a variants containing ncAAs in the acidic pocket display markedly reduced proton sensitivity, with $pH_{50}$ values as low as 5.49 ± 0.13 (T239Bpa, mean ± standard deviation (SD), $n = 6$) and 5.66 ± 0.26 (D357AzF, mean ± SD, $n = 10$). Additionally, we observed substantial changes in current shape compared to WT. For example, current decay rates were increased for T239Bpa (t½ 224 ± 176 ms, $n = 6$) and D357AzF (t½ 93.8 ± 40.9 ms, $n = 10$) compared to WT (t½ 818 ± 750 ms, $n = 9$), indicating possible effects of the photocrosslinkers on channel gating (rates of desensitization or closure; Fig 3B, S3 Table). Overall, we found that incorporation of Se-AbK was least efficient, so all subsequent experiments focused on AzF- and Bpa-containing channel variants.

As hASIC1a variants with ncAAs around the acidic pocket displayed markedly altered proton sensitivity and current decay rates, we next wanted to assess if these variants can still be modulated by 2 peptide gating modifiers that interact with the acidic pocket, BigDyn, and PcTx1.

## Peptide modulation is retained in hASIC1a variants containing photocrosslinkers in the acidic pocket

First, we investigated the neuropeptide BigDyn, which interacts with the acidic pocket and shifts the proton dependence of both activation and steady-state desensitization (SSD) [17]. A key physiological function of BigDyn is to limit ASIC1a SSD [40]. In order to define the appropriate pH for the BigDyn application on each variant, we first established an APC-based protocol to determine SSD curves. Due to the open-well system of the SyncroPatch 384PE, lowering the conditioning pH to assess SSD required multiple mixing steps, which we simulated on a pH meter to determine the apparent pH the cells are exposed to before each activation (S9 Fig). Using this approach, we obtained a $pH_{50}$ SSD of 6.91 ± 0.02 for hASIC1a WT ($n = 40$), which is lower than the value reported in *Xenopus laevis* oocytes ($pH_{50}$ SSD = 7.05 ± 0.01, S10A and S10B Fig). Notably, we also observed a more shallow Hill slope for WT compared to oocytes ($n_H$ 3.16 ± 0.42 versus 9.45 ± 2.84), but not for any of the tested variants in the acidic pocket or interface region (S10B–S10F Fig, S4 Table). SSD profiles of the ncAA-containing variants varied with $pH_{50}$ SSD values ranging from 7.15 ± 0.01 (E177Bpa, $n = 12$) to 6.76 ± 0.06 (K356AzF, $n = 7$, S4 Table), with most variants displaying a slightly increased proton sensitivity compared to WT. This is in contrast to the observed pattern of reduced proton sensitivity for proton-gated activation, suggesting that incorporation of ncAA photocrosslinkers in the acidic pocket modulates proton sensitivity of activation and SSD differentially. For our subsequent APC experiments to assess BigDyn modulation, we chose a conditioning pH that led to around 10% remaining current upon activation.

Here, we focused on AzF-containing variants for which we had previously detected crosslinking to BigDyn on western blots to evaluate if the observed peptide–channel interaction also results in functional modulation [17]. Cells were exposed to SSD-inducing pH conditions in the presence or absence of 3 μM BigDyn, and the resulting currents upon pH 5.6 activation were normalized to control currents after incubation at pH 7.6 (Fig 4A and 4B). Control cells not exposed to BigDyn exhibited SSD to 0% to 30% mean remaining current (S11 Fig, S5 Table), while the BigDyn co-application during conditioning limited SSD to varying degrees (Fig 4B). BigDyn increased rescue from pH-induced SSD in all tested AzF-containing variants, but did not do so in WT, despite a similar trend (Fig 4B). For all tested variants, we regularly

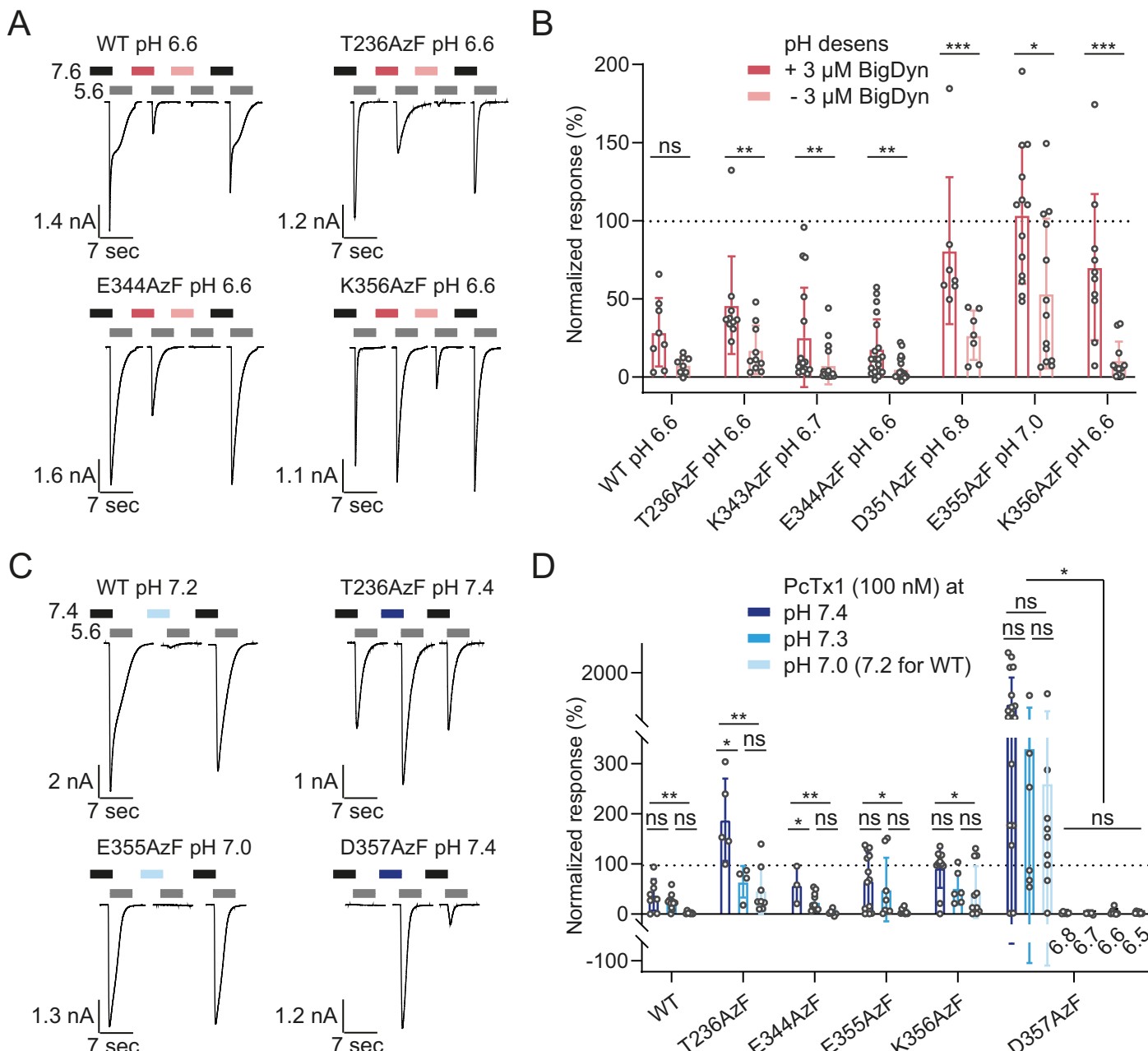

**Fig 4. Peptide modulation of hASIC1a WT and selected variants containing AzF in the acidic pocket. (A)** Characteristic current traces and **(B)** normalized response after SSD in absence or presence of BigDyn for hASIC1a WT and 6 ncAA variants. Cells were incubated at the desensitizing pH specified for each variant with or without 3 μM BigDyn for 2 minutes (pink bars) before activation at pH 5.6 (gray bars, 5 seconds), and the currents were normalized to the average of 2 control currents after conditioning at pH 7.6 (black bars; control traces shown in S11 Fig). **(C)** Exemplary current traces and **(D)** bar graph for PcTx1 modulation of hASIC1a WT and selected variants containing AzF in the acidic pocket at different pH. Cells were incubated with 100 nM PcTx1 at varying pH for 2 minutes (blue bars) before activation at pH 5.6 (gray bars, 5 seconds), and the current was normalized to the average of the 4 preceding and following control currents after conditioning at pH 7.4 (black bars). Bar graphs show mean ± SD, dashed line indicates 100%, and values are shown in S5 and S6 Tables. (*) denotes significant difference between groups, $p < 0.05$; (**): $p < 0.01$; (***): $p < 0.001$; ns: not significant; Mann–Whitney test (B) or 1-way ANOVA with Tukey multiple comparisons test (D). Colored and black bars in (A) and (C) not to scale. The underlying data have been deposited at zenodo.org (https://doi.org/10.5281/zenodo.4906985; files 11, 12, 34, and 35). AzF, 4-Azido-l-phenylalanine; BigDyn, big dynorphin; hASIC1a, human acid-sensing ion channel 1a; ncAA, noncanonical amino acid; PcTx1, psalmotoxin 1; SD, standard deviation; SSD, steady-state desensitization; WT, wild type.

observed incomplete SSD after the first conditioning step, but this typically increased after the second conditioning step (S11 Fig). This could point toward possible confounding effects by the repeated solution mixing to achieve the desired conditioning pH described above. However, despite the reduced control over the conditioning pH compared to using a perfusion system with continuous flow, it was still possible to determine if BigDyn modulates hASIC1a SSD. In short, the APC setup enables rapid evaluation of several channel variants with different SSD profiles for BigDyn modulation in a single experiment.

We next tested a subset of AzF-containing acidic pocket variants for modulation by the gating modifier PcTx1, which was originally isolated from the venom of the *Psalmopoeus cambridgei* tarantula [41]. PcTx1 has previously been shown to increase the apparent proton affinity of both activation and SSD of ASIC1a, resulting in inhibition or potentiation, depending on the application pH [39,41,53,54]. Here, we assessed hASIC1a modulation by co-applying 100 nM PcTx1 at varying conditioning pH and compared the resulting current upon activation with pH 5.6 to the average of the preceding and following control currents after conditioning at pH 7.4 (Fig 4C). For hASIC1a WT, we observed increasing inhibition from $38.2 \pm 31.7\%$ of current remaining at pH 7.4 to $2.06 \pm 2.50\%$ at pH 7.2 (Fig 4D, S6 Table). This is in agreement with previous findings that the PcTx1 $IC_{50}$ decreases at lower pH values [39]. Channel variants with AzF in positions 344, 355, or 356 showed a similar trend (Fig 4D). In contrast, we saw potentiation for T236AzF at pH 7.4 and for D357AzF at pH 7.4 to 7.0 (Fig 4C and 4D). This is consistent with the observation that these variants are among those with the most pronounced reduction in the $pH_{50}$ of activation (Fig 3, S8 Fig, S1 Table, $pH_{50}$ $6.17 \pm 0.14$ ($n = 10$) and $5.66 \pm 0.26$ ($n = 10$), respectively). D357AzF in particular exhibited an unusual phenotype: The first 2 control applications of pH 5.6 led to very small or no detectable channel activation, but pH 5.6 after pre-application of the toxin induced a substantial inward current, after which the channels also activated in response to the following control applications. We therefore chose to evaluate PcTx1 modulation of D357AzF in more detail. Specifically, we used lower pH during conditioning and observed that at pH 6.8 and below, the variant is inhibited. In light of the strong potentiation at pH 7.0, this highlights that PcTx1 modulation of D357AzF exhibits a striking pH dependence, which far exceeds that of WT and the other mutants examined here (Fig 4D, S6 Table) [39].

Overall, the APC assay established here enabled the time-efficient characterization of pharmacological modulation of selected hASIC1a variants, providing an overview on their PcTx1 modulation profile at different application pH. These results confirm that hASIC1a variants containing ncAA photocrosslinkers in the acidic pocket can still be modulated by known peptide gating modifiers, opening avenues to efficiently study peptide–channel interactions with a combination of APC and photocrosslinking.

## Photocrosslinking confirms PcTx1 binding to the hASIC1a acidic pocket

Nine out of the originally targeted 12 positions around the PcTx1 binding site exhibited specific AzF incorporation (Fig 5A, left inset) and were used for photocrosslinking experiments followed by western blotting (workflow in Fig 5B). In parallel, 6 positions in the lower ECD, F69, Y71, V80, D253, W287, and E413 were also replaced by AzF to confirm the specificity of potential photocrosslinking around the acidic pocket. (Fig 5A, right insets). hASIC1a variants were expressed in HEK293T ASIC-KO cells, and 100 nM biotinylated PcTx1 was added before cells were exposed to UV light (365 nm) for 15 minutes to induce photocrosslinking. We then isolated full-length hASIC1a via a carboxyl-terminal 1D4-tag and analyzed protein samples on a western blot with antibodies against biotin and the 1D4-tag to detect PcTx1 and hASIC1a, respectively. Biotinylated PcTx1 was absent in UV-exposed hASIC1a WT and in all control

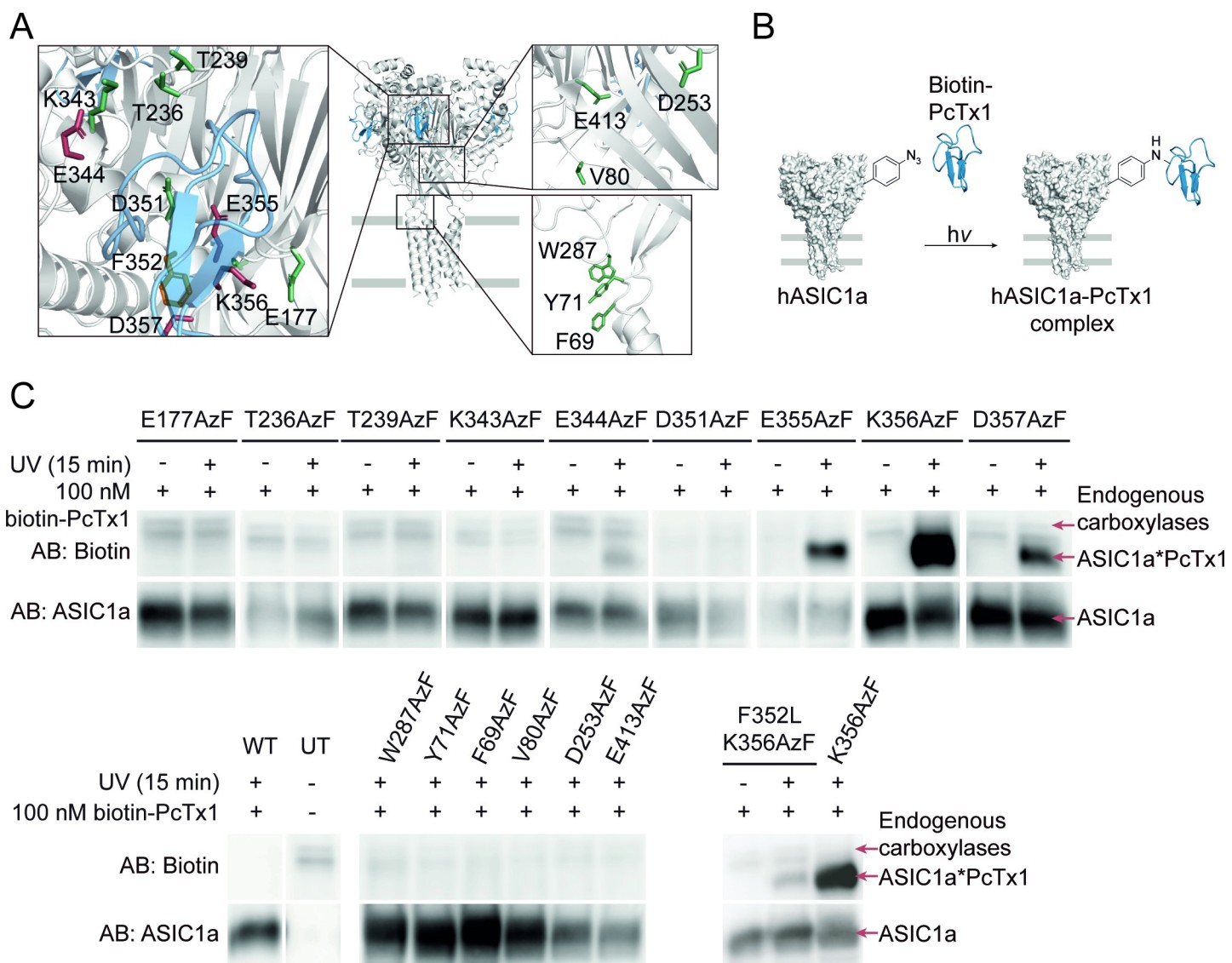

**Fig 5. Live-cell photocrosslinking delineates the PcTx1 binding site at the ASIC1a acidic pocket.** (A) Structure of cASIC1 (white) in complex with PcTx (blue, PDB: 4FZ0), insets show individual side chains replaced by AzF in the acidic pocket (left inset) and lower ECD (right insets). Positions that crosslinked to biotin-PcTx1 are colored red, F352 is marked in orange, and positions that did not crosslink are colored green. (B) Schematic workflow for crosslinking to biotin-PcTx1. HEK 293T ASIC1a-KO cells expressing AzF-containing hASIC1a variants are incubated with 100 nM biotin-PcTx1 and exposed to UV light for 15 minutes to form covalent hASIC1a–PcTx1 complexes, which are purified via a carboxyl-terminal 1D4-tag on hASIC1a and visualized via western blotting. (C) Western blot of purified hASIC1a WT, UT cells, and variants carrying AzF in the ECD detected using the specified AB. Biotin-PcTx1 is detected in UV-exposed samples containing AzF at positions 344, 355, 356, and 357 in the acidic pocket (colored red in A, left inset), but not at positions 177, 236, 239, 343, or 351 (colored green in A, left inset). PcTx1 is also absent in control samples not exposed to UV, those carrying AzF in the lower ECD (right insets in A), WT, or UTs. PcTx1 can be detected upon UV-exposing the toxin-insensitive F352L K356AzF double mutant (left inset in A, F352 colored orange). Of note, the anti-biotin AB detects endogenous biotin-dependent carboxylases, which are also found in purified samples from UTs and have been described before [55,56]. Data are representative of 3 individual experiments; see S12–S15 Figs for original blots and crosslinking attempts with Bpa. The underlying data have been deposited at zenodo.org (https://doi.org/10.5281/zenodo.4906985; files 12 and 13). AB, antibodies; ASIC1a, acid-sensing ion channel 1a; AzF, 4-Azido-l-phenylalanine; Bpa, 4-Benzoyl-l-phenylalanine; cASIC1, chicken acid-sensing ion channel 1; ECD, extracellular domain; hASIC1a, human acid-sensing ion channel 1a; PcTx1, psalmotoxin 1; PDB, Protein Data Bank; UT, untransfected; WT, wild type.

positions containing AzF in the lower ECD (F69, Y71, V80, D253, W287, and E413), as well as in samples containing AzF in the acidic pocket not exposed to UV light (Fig 5C). By contrast, PcTx1 was detected at 4 out of 9 AzF-containing positions (344, 355, 356, and 357) after UV

exposure, indicating covalent photocrosslinking at these positions (marked in red in Fig 5A), but at none of the 5 other sites in the acidic pocket tested (marked in green).

Previous studies have shown that the F352L mutation at the base of the acidic pocket eliminates the modulatory effect of PcTx1 on hASIC1a [57,58], but it remained unclear if the toxin is still able to bind to hASIC1a. To test this possibility directly, we combined the F352L mutation with one of the crosslinking variants, resulting in the hASIC1a F352L K356AzF double mutant variant. Upon UV exposure, we were able to detect the PcTx1–hASIC1a complex even in the presence of the F352L mutation, albeit in lower amounts as assessed by the lower band intensity compared to the K356AzF single variant (Fig 5C, lower panel). This suggests that the F352L mutation does not eliminate toxin binding per se, but likely primarily abolishes the functional effects caused by PcTx1.

Attempts to photocrosslink PcTx1 using Bpa in the equivalent positions around the acidic pocket did not succeed (S12 Fig). We therefore tested PcTx1 modulation of selected Bpa variants on the SyncroPatch 384PE to assess if the toxin binds to the acidic pocket when Bpa is present (S13 Fig, S6 Table). We observed robust inhibition at pH 7.0, indicating the interaction persists despite incorporating a bulky ncAA within the acidic pocket. However, most of the variants showed only weak modulation at the pH used during the UV exposure (7.4), which might partly explain the lack of crosslinking with Bpa.

Overall, our photocrosslinking experiments confirm that PcTx1 interacts with the acidic pocket of hASIC1a, even in the presence of a mutation that abolishes the functional effects of PcTx1.

## Discussion

### First comprehensive functional assessment of ncAA-containing ion channels on an APC platform

Since their introduction, APC platforms have greatly aided ion channel research with their high-throughput capabilities [59]. However, the requirement for high transfection rates to express the ion channels of interest limits the types of experiments that can be performed with this approach. Our FACS-assisted ncAA incorporation assay represents, to our knowledge, the first example of using an APC platform to functionally interrogate ncAA-containing ion channels. By transiently transfecting the protein of interest into mammalian cells and selecting those that express all components with FACS, we circumvent the need for stable cell lines. This method therefore greatly expands the scope of experiments that can be addressed using APC-based approaches.

Our extensive scanning of 309 ncAA-containing variants emphasizes the amenability of hASIC1a to ncAA incorporation, with the highest tolerance observed for AzF (61% functional variants) followed by Bpa (50%) and Se-AbK (44%) (Fig 2E). Previous studies on incorporation of AzF and Bpa into the human serotonin transporter (hSERT) and α-amino-3-hydroxy-5-methyl-4-isoxazolepropionic acid receptor (AMPAR) also show preferred functional incorporation of AzF and attribute this to its smaller size [7,13]. Rannversson and colleagues report lowest ncAA tolerance in the hSERT TMD (44% and 20% for AzF and Bpa, respectively), contrasting our findings in the TM segments of hASIC1a (52% and 61%). However, it should be noted that we specifically selected the outer turns of the TM helices, where the study on AMPARs observed better incorporation compared to the more tightly packed central pore [7].

Previous work on hSERT shows higher success rates for replacing aromatic versus nonaromatic side chains, a trend we only observe for Bpa. Generally, genetic encoding of ncAAs does not appear to depend on the original properties of the replaced amino acid when assessed via

protein expression [14,60]. Indeed, a systematic examination of the effect of the similarly bulky ncAA acridonylalanine on protein solubility found no correlation to amino acid conservation, hydrophobicity, or accessibility, but a close dependence on the location within the overall tertiary structure [61]. Consequently, the authors suggest that scientists broaden rather than narrow screens when aiming to introduce an ncAA into a new target protein. In the present study, we cover around 20% of hASIC1a and functionally assess 3 different ncAAs, likely the most comprehensive investigation of genetic code expansion in a transmembrane protein to date.

## Mechanistic insights into ASIC function

A beneficial side effect of replacing native side chains with ncAA photocrosslinkers is that, in addition to their photoactivatable properties, these bulky side chains can also inform on basic biophysical aspects of the protein domain in question. Here, we show that incorporation of bulky, nonpolar side chains leads to functional channels in about 50% of cases, and we observe a general trend toward lower apparent proton affinity in the ncAA-bearing hASIC1a channels. This is particularly evident at positions in or near the acidic pocket, where previous studies have shown that mutations to acidic side chains in thumb and finger domains result in increased $pH_{50}$ values (reviewed in [23]). By contrast, we only found a few positions in M1 (L45, Q66, and F69) that resulted in higher apparent proton affinity. This is consistent with previous work on the nearby pre-M1 region [62], as well as a number of M1 and M2 mutations that mostly resulted in left-shifted $pH_{50}$ values [48,63]. Together, this suggests that mutations in M1 and M2 of ASIC1a have a tendency to increase apparent proton affinity.

Generally, we observe that the time course of current decay is relatively heterogeneous (Fig 2 and 3, S7 Fig), likely due to the slow and incomplete solution exchange (see also below). This makes an exact quantification of the changes to activation or desensitization rates difficult. Nevertheless, we observe that the same sites around the acidic pocket that show a pronounced decrease in apparent proton affinity also display a marked acceleration in current decay rates ($t_{1/2}$ analysis in Fig 3B, S8 Fig, and S3 Table). This was consistently observed at all of the 8 sites around the acidic pocket assessed in Fig 3 and was independent of the nature of the incorporated ncAA. This finding is coherent with a previous study that showed that the thumb domain affects rates of fast desensitization [64]. Alternatively, it is conceivable that the observed phenotype is due to greatly increased channel deactivation rates [65]. Although we cannot discriminate between these possibilities, our data clearly show that the physicochemical properties of side chains lining the acidic pocket are a major determinant for current decay in ASIC1a.

We also noticed varying degrees of tachyphylaxis, especially when positions in the external turns of the TM helices were replaced with ncAAs (S3 Fig, S1 Table). In light of previous work suggesting a contribution by permeating protons and an effect of hydrophobicity of TM1 side chains on tachyphylaxis, this warrants further investigation [49,66].

## Complex pharmacological modulation studied in ncAA-containing channels using APC

The complex pharmacological modulation pattern of hASIC1a by BigDyn and PcTx1 is notoriously challenging to study. However, we were able to optimize the APC protocols to replicate and even expand on the differential effects of this highly state-dependent peptide modulation (Fig 4). Specifically, we were able to show that despite the prominently lowered proton sensitivity of acidic pocket variants, all tested ncAA-containing hASIC1a variants retained some degree of modulation by both BigDyn and PcTx1. We observed varying degrees of BigDyn-dependent rescue from SSD for the different variants (Fig 4B). Under our conditions, rescue from SSD was incomplete when we applied 3 μM BigDyn, a concentration well above the

reported EC$_{50}$ range of 26 to 210 nM [17,40]. In combination with the steep pH dependence of modulation, this resulted in considerable variability in the BigDyn modulation data, as evident by the reported range in SD values. While this can, at least in part, be attributed to our limited control over the BigDyn application pH, we have made similar observations in a previous study using TEVC [17].

PcTx1 inhibited or potentiated AzF-containing hASIC1a variants in a pH-dependent manner, in line with previous reports [39]. We examined a total of 5 variants, of which all, except T236AzF also formed covalent complexes with the toxin upon UV exposure (Fig 5C). While PcTx1 still modulates and therefore interacts with hASIC1a T236AzF (Fig 4C and 4D), we cannot exclude that introduction of AzF at positions 177, 239, 343, or 351 prevents toxin interaction, as these variants were not assessed for PcTx1 modulation with APC and did not crosslink to the peptide upon UV exposure (Fig 5C).

## Live-cell crosslinking provides a detailed map of the PcTx1–hASIC1a interaction

The acidic pocket is now well established both as a hotspot for channel activation and as a binding site for pharmacological modulators [17,23]. In the case of PcTx1, structural data had already outlined the toxin binding site on ASICs [52,67], but unlike previous work, the crosslinking approach outlined in this study enables us to covalently trap ligand–channel complexes in living cells. This represents a notable advantage, especially for highly state-dependent interactions, such as those between hASIC1a and BigDyn or PcTx1. Additionally, comparing the crosslinking pattern between 2 ligands, the approach can indirectly inform on the varying degrees of conformational flexibility of the ligands: BigDyn is likely to be highly flexible without a strong propensity to adopt a secondary fold [68,69], therefore samples a greater conformational space and is thus more likely to undergo covalent crosslinking at multiple sites (9/9 sites tested at the acidic pocket [17]). By contrast, PcTx1 folds into a compact and highly stable conformation and will consequently undergo covalent crosslinking at relatively fewer sites (4/9 sites tested at the acidic pocket; Fig 5). These findings also complement an earlier investigation of the key interactions between PcTx1 and ASIC1a that concluded for the majority of contacts observed in the crystal structures to not persist during MD simulations or to not be functionally relevant for PcTx1-mediated inhibition of ASIC1a [58].

The ability to covalently trap ligand–receptor complexes offers a unique opportunity to directly assess if ASIC mutations shown to alter or abolish ligand effects still bind to the same site on the receptor. For example, the hASIC1a F352L mutation at the base of the acidic pocket is known to almost completely abolish the PcTx1-dependent modulation of ASIC1a channels [57,58]. Yet it remained unclear if the toxin also interacts with the acidic pocket in these mutant channels. Here, we directly demonstrate that PcTx1 still binds to the acidic pocket, even at a concentration that is far too low to have a functional effect on the mutant channels (100 nM). This leads us to propose that the F352L mutation primarily affects conformational changes responsible for the PcTx1 effect on WT hASIC1a, but not toxin binding per se.

We note that unlike AzF, we were unable to employ Bpa for crosslinking experiments with PcTx1. To test if introduction of the more bulky photocrosslinker prevents toxin interaction, we assessed PcTx1 modulation of selected variants with APC and found robust inhibition for most variants, indicating that Bpa does not fully occlude the acidic pocket (S13 Fig). We therefore speculate that steric constraints due to the positioning of the benzophenone diradical and the more selective reactivity of Bpa (reacts exclusively with C–H bonds) may play a role [70,71]. Together, this emphasizes that screens with multiple redundant ncAAs significantly increase chances of observing successful crosslinking.

## Limitations of the outlined APC-based approach

While our work establishes that ncAA-containing ion channels can be screened on an APC platform, some limitations persist. Firstly, our present approach relies on simultaneous transfection of 4 plasmids (Fig 1), which can negatively impact transfection efficiency and/or result in cells not containing all 4 components. Careful optimization of DNA amounts and transfection conditions is therefore necessary, and a revised construct design to reduce the number of plasmids could further improve yields. For example, the Plested group achieved co-expression of TAG-containing AMPAR and GFP with an internal ribosome entry site (IRES) [5,7], while Rook and colleagues used ASIC1a with a carboxyl-terminally fused GFP tag [8]. Furthermore, both Zhu and colleagues and Rook and colleagues created bidirectional plasmids to encode both AzF-RS or Bpa-RS and tRNA, respectively [6,8]. This latter strategy might be particularly fruitful for the incorporation of Se-AbK, which was generally less efficient than that of AzF and Bpa (Fig 2E, S1 Table), despite others reporting robust incorporation of a similar ncAA [72].

Secondly, while GFP fluorescence indicates successful transfection and ncAA incorporation and thereby increased likelihood of observing proton-gated currents in cells grown in the presence of ncAA, it is not a reliable proxy for incorporation specificity in control cells grown in the absence of ncAA. This is due to the fact that the degree of unspecific incorporation in GFP does not correlate with that of the ncAA-containing hASIC1a variants. We consistently observed GFP fluorescence in only around 2% of the control cells, independent of the co-expressed channel variant, which translated to insufficient cell numbers for APC (requires a minimal concentration of 100,000 cells/ml). Assuming that the transfection rates are similar in the presence and absence of ncAA (i.e., around 11%, S1B Fig), we concluded that recording a larger number of unsorted control cells is the more stringent approach to assess incorporation efficiency. We therefore did not subject the incorporation control cells to FACS and instead conducted APC with the entire unsorted cell population. To evaluate this strategy, we randomly selected 45 hASIC1a variants assessed for ncAA incorporation in the N-terminus, ECD, or carboxyl terminus and compared the number of wells harboring a patched cell with >100 MΩ seal and those showing proton-gated currents in presence and absence of ncAA (S1C Fig). While the percentage of cells with current is generally lower for cells grown in the absence of ncAA, we do observe currents for those positions where incorporation is unspecific, e.g., throughout the carboxyl terminus and in some positions in the N-terminus. For the control samples, an average of 9.8 out of 16 possible wells contained a cell with >100 MΩ seal, and we observed currents in 1.6 wells on average. We therefore conclude that despite some shortcomings, the employed strategy using nonsorted controls detects at least those positions with unspecific incorporation of >15% (i.e., 1.6/9.8).

Thirdly, while APC platforms offer unprecedented throughput and speed, there are limitations with regard to the rate and extent of perfusion exchange. This can be particularly challenging for ligand application to fast-gating ligand-gated ion channels (i.e., pH changes for ASIC1a) in general and strongly state-dependent pharmacological modulation (by e.g., Big-Dyn or PcTx1) in particular. Although we were able to partially overcome these issues by employing a solution stacking approach, we cannot draw detailed conclusions about activation or desensitization kinetics. Similarly, values for proton-dependent activation and especially SSD can be determined with greater precision using TEVC or manual patch clamp electrophysiology. However, note that the values reported here are generally in agreement with previous reports, both with regard to WT values [46,47] and relative shifts caused by mutations, i.e., in the acidic pocket [23].

Lastly, limitations arise from the accessibility and running costs of APC platforms compared to conventional patch clamp setups. But we anticipate that the establishment of

academic core facilities for high-throughput electrophysiology (e.g., Northwestern University, Il, US; University of Nantes, France; and Illawarra Health and Medical Research Institute, Wollongong, Australia) and collaborations between academia and industry (this study, [73–76]) will likely contribute to a broader accessibility. This is also evident from the rising number of publications involving APC (currently >80 publications according to vendor information).

## Conclusions and outlook

The ability to functionally screen ncAA-containing ion channels on APC platforms has the potential to greatly expand the use of ncAAs in both academic and industry settings. The intrinsically high throughput enables rapid assessment of incorporation efficiencies, functional properties, and even complex pharmacological modulation. In principle, the approach can be used for both site-specific (this study) and global ncAA incorporation [77,78], thus further increasing the number and type of chemical modifications that can be introduced. In the case of incorporation of photocrosslinking ncAAs, the approach can be exploited to crosslink to peptides (Fig 5 [17]), small molecules [13], or establish intraprotein crosslinking, including in protein complexes [8,9]. Furthermore, the recently developed ability for on-chip optostimulation on related APC platforms [79] offers exciting prospects for potentially conducting UV-mediated crosslinking during live APC experiments in the future. Paired with mass spectrometry and/or biochemical approaches [80,81], the overall strategy could also be expanded to define interaction sites of unknown or known protein–protein interactions. Given that there are now well over 100 different ncAAs available for incorporation into proteins in mammalian cells [1,82], the above approach will enable the efficient study of ion channels endowed with a wide range of properties or functionalities.

## Materials and methods

### Molecular biology

The complementary DNA (cDNA) encoding hASIC1a was kindly provided by Dr. Stephan Kellenberger. Plasmids containing AzF-RS, Bpa-RS, and tRNA were gifts from Dr. Thomas P. Sakmar [43]. AbK-RS and tRNA$_{pyl}$ in pcDNA3.1 were kindly provided by Dr. Chris Ahern [44]. The dominant negative eukaryotic release factor (DN-eRF) was a gift from Dr. William Zagotta [83]. Plasmids containing rat GluA2 Q607 Y533TAG or S729TAG were gifts from Dr. Andrew Plested [5,7], and rat GluA2 Q607 WT was kindly provided by Dr. Anders Skov Kristensen. Rat P2X2 WT 3T was a gift from Dr. Thomas Grutter [50]; the K296TAG variant was generated in-house.

Site-directed mutagenesis was performed using PfuUltraII Fusion polymerase (Agilent, Denmark) and custom DNA mutagenesis primers (Eurofins Genomics, Germany). All sequences were confirmed by sequencing of the full coding frame (Eurofins Genomics). For hASIC1a constructs, a carboxyl-terminal 1D4-tag was added for protein purification and western blot detection, and 2 silent mutations were inserted at V10 and L30 to reduce the risk of potential reinitiation [84].

### Cell culture and transfection

HEK 293T cells (ATCC, Virginia, United States of America), in which endogenous hASIC1a was removed by CRISPR/Cas9 [17], were grown in monolayer in T75 or T175 flasks (Orange Scientific, Belgium) in DMEM (Gibco, Denmark) supplemented with 10% FBS (Thermo Fisher Scientific, Denmark) and 1% penicillin-streptomycin (Thermo Fisher Scientific) and incubated at 37˚C in a humidified 5% $CO_2$ atmosphere. For APC experiments, cells were

seeded into 6-well plates (Orange Scientific) at a density of 300,000 cells/well and transfected the next day with Trans-IT LT1 (Mirus, Wisconsin, United States of America) and 1:1:1:1 μg DNA encoding hASIC1a TAG variants, ncAA-RS, tRNA, and eGFP Y40TAG or Y151TAG, respectively. For the WT control, cells were transfected with 1 μg hASIC1a WT and 0.3 μg eGFP WT. Six hours after transfection, cell medium was replaced with supplemented DMEM containing 10 μM AzF- or Bpa-methylester (for synthesis and mass spectrometric incorporation analysis, see S1 Text) or 100 μM Se-AbK (custom-synthesized by ChiroBlock, Germany). FACS and APC recordings were performed 48 hours after transfection. The same procedure was used for GluA2 and P2X2R recordings.

For crosslinking studies, cells were seeded into 15-cm dishes (VWR, Denmark) at a density of 5 to 7 million cells and transfected the next day with PEI (Polysciences, Germany) and 16:4:4:8 μg DNA encoding hASIC1a TAG variants, AzF-RS, tRNA, and DN-eRF, respectively. For WT controls, 2 million cells were seeded into a 10 cm dish (VWR) and transfected with 8 μg hASIC1a WT. Six hours after transfection, cell medium was replaced with supplemented DMEM containing 0.5 mM AzF (Chem-Impex, Illinois, USA) or 1 mM Bpa (Bachem, Switzerland) and crosslinking studies were performed 48 hours after transfection. Please note that for crosslinking studies followed by western blot, the free acid version of the ncAAs was used to increase protein yields.

## FACS

HEK 293T cells were washed with PBS, treated with Accutase (Sigma-Aldrich, Denmark) or Trypsin-EDTA (Thermo Fisher Scientific), pooled and centrifuged at 1,000 rpm for 5 minutes. They were resuspended in 350 μl of a 1:1 mixture of serum-free Hams F-12 nutrient mixture and extracellular patch clamp solution (140 mM NaCl, 4 mM KCl, 1 mM $MgCl_2$, 2 mM $CaCl_2$, 10 mM HEPES, pH 7.4) supplemented with 20 mM HEPES and transported to the FACS Core Facility at ambient temperature. A FACSAria I or III (BD Biosciences, California, USA) with a 70-μm nozzle was used to sort cells for singularity, size and GFP fluorescence (Excitation 488 nm, Emission 502 nm (low pass) and 530/30 nm (band pass)). Cells were filtered through a sterile 50-μm cup filcon (BD Biosciences) directly before sorting to prevent clogging of the nozzle. The WT control was used to set the fluorescence cutoff between GFP-positive and GFP-negative populations and to check the purity of the sort before sorting 1 million GFP-positive cells for subsequent patch clamp experiments. Where possible, a minimum of 200,000 GFP-positive cells were collected for hASIC1a TAG variants grown in presence of ncAA, while controls grown in absence of ncAA and untransfected cells were not sorted. Cells were collected in 1.5-ml tubes containing the 1:1 mixture mentioned above and transported to the APC instrument at ambient temperature.

## Automated patch clamp

Automated whole-cell patch clamp recordings were conducted on a SyncroPatch 384PE (Nanion Technologies) directly after FACS sorting. Cells were loaded into a teflon-coated plastic boat at concentrations of 1 million cells/ml (WT, controls grown in absence of ncAA and untransfected cells) or 200,000 to 400,000 cells/ml (variants grown in presence of ncAA) and incubated at 20°C and 200 rpm. For patch clamp recordings, a NPC-384 medium resistance single hole chip (Nanion Technologies) was filled with intracellular solution (120 mM KF, 20 mM KCl, 10 mM HEPES, pH 7.2) and extracellular solution (140 mM NaCl, 4 mM KCl, 1 mM $MgCl_2$, 2 mM $CaCl_2$, 10 mM HEPES, pH 7.4). A total of 30 μl of cells were loaded into each well, and the cells were caught on the holes by brief application of −200 mbar pressure and washed with 30 μl seal enhancer solution (extracellular solution with 8 mM $Ca^{2+}$) under a

holding pressure of −50 mbar. After a wash step with extracellular solution, 2 more pulses of −200 mbar were applied to reach whole cell configuration, and the cells were clamped at 0 mV under atmospheric pressure (S9A Fig). For recordings of concentration response curves, extracellular solutions at different pH were applied using a liquid stacking approach. Briefly, pipette tips were loaded with 45 μl of pH 7.4 wash solution followed by 5 μl of activating extracellular solution (pH 7.2 to 4.8). For each sweep, the baseline current was recorded for 1 second before the application of the 5 μl activating solution, while the pH 7.4 wash solution was dispensed with a delay of 5 seconds to allow for recording of channel opening and desensitization in the presence of ligand. The second dispension was directly followed by aspiration of liquid and a second wash step with pH 7.4 before the application of the next activating pH (interval between stimuli 140 seconds, S9B Fig).

For SSD curve recordings, cells were exposed to an activating pH of 5.6 using the stacked addition protocol described above, while the conditioning pH was varied (pH 7.6 to 6.4). The open-well system of the SyncroPatch 384PE does not allow a single exchange of the entire liquid surrounding the cell, as this would result in destabilization or loss of the seal. Instead, the conditioning pH was adjusted stepwise by repeated addition and removal of 50% of the solution in the well, leading to 6 minutes conditioning intervals between stimuli (S9C Fig). While this process was simulated at the pH meter to determine the apparent conditioning pH, small variations may occur due to mixing effects. The authors note that APC instruments operating with microfluidic flow channels might offer superior control of the conditioning pH. At the end of each SSD curve recording, a control application of pH 5.6 after conditioning pH 7.6 was used to assess the extent of current rescue and exclude cells that did not recover from SSD.

For peptide modulation experiments, 0.1% (w/v) bovine serum albumin (BSA, Sigma-Aldrich) was added to the conditioning solutions to reduce peptide loss on boat and tip surfaces. To investigate modulation by BigDyn (synthesis described in [17]), cells were first exposed to 2 activations with pH 5.6 after conditioning at pH 7.6 to determine the control current, followed by 2 rounds of activation after 2 minutes conditioning with a pH that induces SSD (total interval between stimuli: 8 minutes, due to the conditioning protocol described above) and a control activation to evaluate current recovery. For half of the cell population, 3 μM BigDyn were co-applied during the second conditioning period to measure rescue from SSD. This assessment of SSD and recovery was repeated with peptide co-application during the first SSD conditioning to also evaluate peptide wash out. To assess modulation by PcTx1 (Alomone Labs, Israel, >95% purity), cells were exposed to 2 control measurements of activation with pH 5.6 after conditioning at pH 7.4 (interval 3.75 minutes), followed by pH 5.6 activation after incubation with 100 nM PcTx1 at varying pH (pH 7.4 to 7.0) for 2 minutes (total interval between stimuli 7 minutes), as well as 2 further controls to assess recovery from modulation.

For recordings on GluA2 and P2X2R variants, cells were clamped at −60 mV, and currents activated by application of 30 μM and 300 μM/10 mM ATP or glutamate, respectively. Cells expressing GluA2 were preincubated with 100 μM cyclothiazide (in 0.8% (v/v) DMSO) for 60 seconds before activation to slow desensitization (total interval between stimuli: 220 seconds) [51].

## Data analysis

Current traces were acquired at 2 kHz and filtered in the DataControl384 software using a Butterworth 4th order low-pass filter at 45 Hz to remove solution artifacts. Only cells with initial seals >100 MΩ were considered for biophysical characterization using GraphPad Prism 7 or 8, while wells with lower seals, no current, or no caught cell were excluded. The relatively low

seal cutoff in combination with the large proton-gated currents (up to 10 nA) recorded for WT and some of the ncAA-containing variants resulted in suboptimal voltage clamp conditions for a subpopulation of cells, as also apparent from the current shapes. However, we have no evidence that this adversely affected activation parameters or pharmacological modulation. Where possible, APC data were pooled from a minimum of 3 cells and 2 separate recording days. On several occasions, an $n$ of 5 or more was acquired during the first screening trial, in which case the experiment was not repeated. Current sizes were normalized to the respective control currents and half-maximal concentrations (EC$_{50}$ values) and Hill slopes (n$_H$) calculated using Eq (1). pH$_{50}$ values were calculated in Excel using Eq (2). All values are expressed as mean ± SD. The extent of tachyphylaxis for each recording was calculated by subtraction of the normalized current at lowest pH from the normalized maximal current (>20% tachyphylaxis is marked by ($^\#$)). Bar graphs and dot plots were made using GraphPad Prism 7 or 8 and SigmaPlot 13.0, while current traces were exported to Clampfit 10.5 and Adobe Illustrator CC 2019.

$$Y = \frac{100 * (EC_{50}{}^{Hillslope})}{(EC_{50}{}^{Hillslope} + (X^{Hillslope}))} \tag{1}$$

$$\mathrm{pH}_{50} = -\log_{10}(\mathrm{EC}_{50}[\mathrm{M}]) \tag{2}$$

Mean current sizes and pH$_{50}$ values of different cell lines and constructs were compared using Student $t$ test, Mann–Whitney test, or 1-way ANOVA followed by Tukey multiple comparisons test.

## Crosslinking studies, protein purification, and western blotting

Cells were washed with PBS and dislodged using cell scrapers (Orange Scientific). After centrifugation (1,000 rpm, 5 minutes), cell pellets were resuspended in 1 mL PBS pH 7.4 containing 100 nM biotinyl-PcTx1 (Phoenix Pharmaceuticals, California, USA) and transferred into 12 well plates (Orange Scientific). Cells were placed on ice and crosslinked at a distance of 7 to 10 cm to a Maxima ML-3500 S UV-A light source (Spectronics, New York, United States of America, 365 nm) for 15 minutes (AzF) or up to 60 minutes (Bpa). Control samples without UV exposure were kept at 4˚C. After crosslinking, cells were centrifuged (1,000 rpm, 5 minutes) and resuspended in 1 mL solubilization buffer (50 mM Tris-HCl, 145 mM NaCl, 5 mM EDTA, 2 mM DDM, pH 7.5) supplemented with cOmplete EDTA-free protease inhibitor cocktail (Sigma-Aldrich). Cells were lysed (2 hours, 4˚C) and centrifuged for 30 minutes (18,000 g/4˚C). In parallel, 40 μL Dynabeads Protein G (Thermo Fisher Scientific) were washed with 200 μL PBS/0.2 mM DDM and incubated with 4 μg RHO 1D4 antibody (University of British Columbia) in 50 μL PBS/0.2 mM DDM on a ferris wheel (VWR, 30 minutes). After washing the beads with 200 μL PBS/0.2 mM DDM (with 200 μL), the cell lysate supernatant was incubated with the beads on a ferris wheel (4˚C, 90 minutes). Beads were washed with 200 μL PBS 3 times to remove nonspecifically bound proteins and incubated in 25 μL elution buffer (2:1 mixture between 50 mM glycine, pH 2.8 and 62.5 mM Tris-HCl, 2.5% SDS, 10% Glycerol, pH 6.8) supplemented with 80 mM DTT at 70˚C for 10 minutes. Protein samples (12 μL) were mixed with 3 μL 5 M DTT and 5 μL 4x NuPAGE LDS sample buffer (Thermo Fisher Scientific) and incubated (95˚C, 20 minutes) before SDS-PAGE using 3% to 8% Tris-Acetate protein gels (Thermo Fisher Scientific). After transfer onto PVDF membranes (iBlot 2 Dry Blotting System, Thermo Fisher Scientific) and blocking in TBST/3% nonfat dry milk for 1 hour, hASIC1a was detected using RHO 1D4 antibody (1 μg/μL, University of British Columbia) and 1:5,000 goat anti-mouse IgG HRP-conjugate (Thermo Fisher Scientific).

Biotinyl-PcTx1 was detected using 1:1,000 rabbit anti-biotin antibody (Abcam, United Kingdom) and 1:5,000 goat anti-rabbit IgG HRP-conjugate (Promega, Denmark). Samples used for incorporation controls were treated as above, but were not exposed to UV light.

## Supporting information

**S1 Fig. FACS prior to APC increases current density on the 384-well chip.** **(A)** Transfection efficiency and percentage of wells showing proton-gated currents (of wells harboring a patched cell with >100 MΩ seal) for hASIC1a WT co-transfected with WT GFP, GFP Y151AzF, or GFP Y40Bpa. While the transfection efficiency is reduced considerably from 61.6% to 29.9% and 33.1% upon co-transfection of GFP Y151AzF or GFP Y40Bpa, respectively, the decrease in mean current density is less pronounced from around 48% to 39% and 42%. This illustrates that nonsense suppression in GFP results in a decrease in apparent TE compared to WT and shows that both WT and TAG-containing GFP can be used as a reporter to enrich transfected cells for APC. **(B)** Transfection efficiency of 45 randomly selected hASIC1a variants assessed for ncAA incorporation in the N-terminus, ECD or carboxyl terminus. Of note, only cells grown in the presence of ncAA were FACS-sorted and assessed for transfection efficiency. **(C)** Percentage of wells showing proton-gated currents (of wells harboring a patched cell with >100 MΩ seal). Cells grown in the absence of ncAA (lighter shades in right panel) show currents in 0%–50% of the wells, depending on incorporation specificity, which varies in the different protein domains. All values shown as mean ± SD. The underlying data have been deposited at zenodo.org (https://doi.org/10.5281/zenodo.4906985; file 14). APC, automated patch clamp; ECD, extracellular domain; FACS, fluorescence-activated cell sorting; hASIC1a, human acid-sensing ion channel 1a; ncAA, noncanonical amino acid; SD, standard deviation; TE, transfection efficiency; WT, wild type.
(TIF)

**S2 Fig. Incorporation of ncAA photocrosslinkers into the hASIC1a N-terminus is specific from position 10 onward and mostly produces variants with WT-like properties.** **(A)** Snake plots with tested positions marked in blue. **(B)** Dot plots comparing peak current sizes (left) and $pH_{50}$ (right); bars indicate mean ± SD, and ([#]) marks >20% tachyphylaxis (see also S1 Table). For variants expressed in the absence of ncAAs that yielded currents, results are marked by underlying gray bars. The underlying data have been deposited at zenodo.org (https://doi.org/10.5281/zenodo.4906985; files 15–18). hASIC1a, human acid-sensing ion channel 1a; ncAA, noncanonical amino acid; SD, standard deviation; WT, wild type.
(DOCX)

**S3 Fig. In the hASIC1a outer transmembrane helix loops and interface region, incorporation of ncAA photocrosslinkers is better tolerated in the M1 than the M2 helix and produces variants with varying degrees of tachyphylaxis.** **(A)** Snake plots with tested positions marked in blue. **(B)** Several variants, e.g., A47AzF, Y68Bpa, G340Se, and Y458AzF, undergo tachyphylaxis after reaching the peak current. **(C)** Dot plots comparing $pH_{50}$ (left) and peak current sizes (right); bars indicate mean ± SD, and ([#]) marks >20% tachyphylaxis (see also S1 Table). For variants expressed in the absence of ncAAs that yielded currents, results are marked by underlying gray bars. The underlying data have been deposited at zenodo.org (https://doi.org/10.5281/zenodo.4906985; files 19–23). hASIC1a, human acid-sensing ion channel 1a; ncAA, noncanonical amino acid; SD, standard deviation.
(DOCX)

**S4 Fig. Incorporation of ncAA photocrosslinkers into the hASIC1a carboxyl terminus is unspecific from position 465 onward.** **(A)** Snake plots with tested positions marked in blue.

**(B)** Dot plots comparing peak current sizes (left) and $pH_{50}$ (right); bars indicate mean ± SD, and (#) marks >20% tachyphylaxis (see also S1 Table). For variants expressed in the absence of ncAAs that yielded currents, results are marked by underlying gray bars. The underlying data have been deposited at zenodo.org (https://doi.org/10.5281/zenodo.4906985; files 24–28). hASIC1a, human acid-sensing ion channel 1a; ncAA, noncanonical amino acid; SD, standard deviation.
(DOCX)

**S5 Fig. hASIC1a carboxyl-terminal positions distal of L465 are not essential for proton-gated current responses. (A)** Snake plot of hASIC1a highlighting carboxyl-terminal positions in blue. **(B)** Representative current traces of C466TAG with and without Bpa (upper panels) and R467TAG with and without Se-AbK (termed "Se," lower panels) as recorded on the SyncroPatch 384PE. **(C)** Dot plots comparing $pH_{50}$ (left) and peak current sizes (right); bars indicate mean ± SD and (#) marks >20% tachyphylaxis (see also S1 Table). For variants expressed in the absence of ncAAs that yielded currents, results are marked by underlying gray bars. **(D)** Concentration response curves of hASIC1a WT (black) and carboxyl-terminally truncated constructs recorded in HEK 293T cells (APC, left panel) and *X. laevis* oocytes (TEVC, right panel). **(E)** Western blot using an ASIC1a-antibody targeting an extracellular epitope documents truncation of the protein. The underlying data have been deposited at zenodo.org (https://doi.org/10.5281/zenodo.4906985; files 29–31). APC, automated patch clamp; Bpa, 4-Benzoyl-l-phenylalanine; hASIC1a, human acid-sensing ion channel 1a; ncAA, noncanonical amino acid; SD, standard deviation; TEVC, two-electrode voltage clamp; WT, wild type.
(TIF)

**S6 Fig. (A)** Incorporation of AzF and Bpa in the carboxyl terminus is efficient with position-dependent specificity. Selected hASIC1a TAG variants were expressed in presence or absence of 10 µM AzF-ME or 1 mM Bpa in HEK 293T ASIC1a-KO cells for 48 hours; the full-length protein was purified via a carboxyl-terminal 1D4-tag and visualized by western blotting using the indicated antibody (AB). Only small amounts of full-length protein were detected in the absence of ncAA, indicating efficient incorporation. **(B)** Mass spectrometry confirms incorporation of Bpa at position 480. HCD fragment ion mass spectrum of the precursor peptide SBpaDKGVAL (positions 479–486, green) and the corresponding fragment ions (a- and b-ion series red, y ion series blue). Theoretical peptide mass 940.478, experimental $m/z$ 470.743 (+-0.53 ppm), charge +2. The underlying data have been deposited at zenodo.org (https://doi.org/10.5281/zenodo.4906985; file 31). AzF, 4-Azido-l-phenylalanine; Bpa, 4-Benzoyl-l-phenylalanine; hASIC1a, human acid-sensing ion channel 1a; HCD, higher-energy C-trap dissociation; ncAA, noncanonical amino acid.
(TIF)

**S7 Fig. The established FACS APC workflow is suitable to assess ncAA incorporation into other ligand-gated ion channels.** Example current traces and dot plots comparing current sizes for WT and ncAA-containing variants of the P2X2 receptor **(A)** and GluA2 **(B)** at different concentrations of ATP or glutamate, respectively. Cells expressing GluA2 were incubated with 100 µM cyclothiazide (0.8% v/v DMSO) for 1 minute before glutamate addition to reduce rapid desensitization [3]. Black arrow indicates ligand application, and gray arrow indicates addition of wash solution. As apparent from the current traces, application of the wash solution removes the ligand from the channels temporarily, but as it is still present in the well, channels can reopen, and desensitize over time. In order to enable concentration response curve measurements, all ligand has to be removed from the well. Bar graphs are mean ± SD, and values are shown in S2 Table. The underlying data have been deposited at zenodo.org

(https://doi.org/10.5281/zenodo.4906985; file 32). APC, automated patch clamp; FACS, fluorescence-activated cell sorting; ncAA, noncanonical amino acid; SD, standard deviation; WT, wild type.
(TIF)

**S8 Fig. Evaluation of 12 positions around the PcTx1 binding site reveals several channel variants with lowered proton sensitivity and accelerated current decay. (A)** Snake plot of hASIC1a highlighting assessed positions in blue. **(B)** Representative current traces of T239Bpa, D351AzF, and D357AzF as recorded on the SyncroPatch 384PE, with arrows indicating time of proton application. Dashed lines indicate WT current in response to pH 6.0 application. **(C)** Dot plots comparing $pH_{50}$ (left) and peak current sizes (right); bars indicate mean ± SD, and (#) marks >20% tachyphylaxis (see also S1 Table). The underlying data have been deposited at zenodo.org (https://doi.org/10.5281/zenodo.4906985; file 33). hASIC1a, human acid-sensing ion channel 1a; PcTx1, psalmotoxin 1; SD, standard deviation; WT, wild type.
(TIF)

**S9 Fig. Schematic representations of the patching process, stacked ligand addition, and exchange of the conditioning pH on the SyncroPatch 384PE. (A)** After filling the wells with external and internal solution (light and dark blue), cells (orange) are added and caught on the hole by brief application of −200 mbar pressure. The cells are held in place with −50 mbar during the wash steps with seal enhancer and external solution before going into whole cell configuration via 2 pulses at −200 mbar. **(B)** For the stacked ligand application, pipettes are filled with 45 μl of resting pH ($pH_{rest}$) followed by 5 μl solution of activating pH ($pH_{act}$). Dispension of $pH_{act}$ leads to channel activation and desensitization in the presence of ligand, followed by dispension of $pH_{rest}$ with a delay of 5 seconds to wash out the ligand. Solution is slowly taken back up into the pipette at the end of each sweep, followed by a wash step with $pH_{rest}$ (not shown). **(C)** When measuring SSD curves, the open-well system of the SyncroPatch 384PE requires repeated mixing steps to approximate the target conditioning pH without disturbing the cell (orange). Moreover, 50% of the liquid (blue) are aspirated and replaced by lower pH solution (gray) twice to obtain the final conditioning pH, here pH 7.15. SSD, steady-state desensitization.
(TIF)

**S10 Fig. SSD of hASIC1a WT and 17 variants containing AzF or Bpa in the acidic pocket or interface region can be efficiently assessed using APC. (A)** Example current trace for hASIC1a WT. Currents were evoked by application of pH 5.6 after conditioning at the indicated pH values for 2 minutes. Current recovery was assessed at the end of the protocol and cells that did not regain current were excluded from the analysis. **(B–F)** SSD curves of WT, 14 variants carrying AzF or Bpa in the acidic pocket, and 3 control positions in the interface region. Currents were normalized to the mean of the first 2 applications. See S4 Table for $pH_{50}$ SSD values, nH and n. The underlying data have been deposited at zenodo.org (https://doi.org/10.5281/zenodo.4906985; files 34 and 35). APC, automated patch clamp; AzF, 4-Azido-l-phenylalanine; Bpa, 4-Benzoyl-l-phenylalanine; hASIC1a, human acid-sensing ion channel 1a; SSD, steady-state desensitization; WT, wild type.
(TIF)

**S11 Fig. BigDyn modulation of hASIC1a WT and 6 variants carrying AzF in the acidic pocket. (A–C)** Characteristic current traces of the full APC protocols for WT (A), T236AzF (B), and E344AzF (C) with and without 3 μM BigDyn (lower versus upper panel). Cells were first exposed to 2 activation pulses with pH 5.6 (gray bars, 5 seconds) after conditioning at pH 7.6 (black bars) to determine the control current, followed by 2 rounds of activation after 2

minutes conditioning with a pH that induces SSD (light pink bars) and a control pulse to evaluate current recovery. For half of the cell population, 3 μM BigDyn (dark pink bars) were co-applied during the second conditioning period to measure rescue from SSD. This assessment of SSD and recovery was repeated with peptide co-application during the first SSD conditioning to also evaluate peptide wash out. Currents were normalized to the average of the first 2 control pulses to compare modulation at different conditions (values in S5 Table, pink and black bars not to scale). **(D)** Bar graph comparing current after SSD in absence and presence of 3 μM BigDyn (traces 3+4 in A–C, lower panel). **(E)** Bar graph comparing current after SSD for control cells not exposed to BigDyn (traces 3+4 and 6+7 in A–C, upper panel). Bar graphs show mean ± SD, dashed line indicates 100%, and values are shown in S5 Table. (*) denotes significant difference between groups, $p < 0.05$; (**): $p < 0.01$; (***): $p < 0.001$; ns: not significant; Mann–Whitney test. The underlying data have been deposited at zenodo.org (https://doi.org/10.5281/zenodo.4906985; file 11). APC, automated patch clamp; AzF, 4-Azido-l-phenylalanine; hASIC1a, human acid-sensing ion channel 1a; SSD, steady-state desensitization; WT, wild type.
(TIF)

**S12 Fig. PcTx1 is not detected in the ASIC1a acidic pocket when Bpa is used for photocrosslinking. (A)** Structure of cASIC1 (white) in complex with PcTx1 (blue, PDB: 4FZ0), inset shows individual side chains replaced by Bpa in the acidic pocket (green), none of which crosslinked to biotin-PcTx1. **(B)** Schematic workflow for Bpa crosslinking to biotin-PcTx1 (see also Fig 5). **(C)** Western blot of purified hASIC1a K356AzF, UT cells, and variants carrying Bpa in the ECD detected using the specified antibodies (AB). Biotin-PcTx1 is only detected in the control sample containing AzF at position 356 (15 minutes UV exposure), but not in any of the 9 positions containing Bpa (60 minutes UV exposure, positions colored green in A). The detected double band by the anti-biotin AB originates from endogenous biotin-dependent carboxylases [4, 5]. **(D)** Control experiments demonstrating efficient Bpa incorporation at all positions tested for crosslinking in C. Stop-codon containing hASIC1a mutants were grown in the presence or absence of 1 mM Bpa in HEK 293T cells for 48 hours, after which the resulting full-length protein was purified via a carboxyl-terminal 1D4-tag and visualized by western blotting using the indicated antibody (AB). With the exception of positions 236 and 357, only small amounts of full-length protein were detected in absence of Bpa (compared to those obtained in its presence), demonstrating efficient incorporation. The underlying data have been deposited at zenodo.org (https://doi.org/10.5281/zenodo.4906985; file 13). AzF, 4-Azido-l-phenylalanine; Bpa, 4-Benzoyl-l-phenylalanine; cASIC1, chicken acid-sensing ion channel 1; ECD, extracellular domain; hASIC1a, human acid-sensing ion channel 1a; PcTx1, psalmotoxin 1; PDB, Protein Data Bank; UT, untransfected.
(TIF)

**S13 Fig. Bar graph for PcTx1 modulation of hASIC1a WT and selected variants containing Bpa in the acidic pocket at different pH.** Cells were incubated with 100 nM PcTx1 at varying conditioning pH for 2 minutes before activation at pH 5.6, and the current was normalized to the average of the 4 preceding and following control currents after conditioning at pH 7.4. Bar graph shows mean ± SD, dashed line indicates 100%, and values are shown in S6 Table. (**) denotes significant difference between groups, $p < 0.01$; ns: not significant; Mann–Whitney test. The underlying data have been deposited at zenodo.org (https://doi.org/10.5281/zenodo.4906985; file 12). Bpa, 4-Benzoyl-l-phenylalanine; hASIC1a, human acid-sensing ion channel 1a; PcTx1, psalmotoxin 1; SD, standard deviation; WT, wild type.
(TIF)

**S14 Fig. Original western blots for AzF crosslinking.** Black bars indicate protein ladders for clarity (left panel), and original markers (coomassie) are overlayed with the blot (chemiluminescence, right panel). Areas cropped for Fig 5 are marked with boxes. **(A)** Western blot for positions 236, 239, 343, 356, and 357. **(B)** Western blot for positions 177, 239, and 344. **(C)** Western blot for positions 71, 287, 69, 80, 253, 413, 351, and 355. **(D)** Western blot for the F352L K356AzF double mutant. Data are representative of 2–3 individual experiments. Control experiments demonstrating efficient AzF incorporation for all above positions are published in [6]. The data have also been deposited at zenodo.org (https://doi.org/10.5281/zenodo.4906985; file 13). AzF, 4-Azido-l-phenylalanine.
(DOCX)

**S15 Fig. Original western blots for Bpa crosslinking.** Black bars indicate protein ladders for clarity (left panel), and original markers (coomassie) are overlayed with the blot (chemiluminescence, right panel). Areas cropped for S12 Fig are marked with boxes. **(A)** Western blot for positions 177, 236, 239, and 343, including K356AzF as a positive control. **(B)** Western blot for positions 344, 351, 355, 356, and 357, including K356AzF as a positive control. **(C)** Western blot demonstrating efficient Bpa incorporation at positions 239, 343, 344, 351, 355, and 357 (upper panel) and at positions 177, 236, and 356 (lower panel). Data are representative of 2–3 individual experiments. The data have also been deposited at zenodo.org (https://doi.org/10.5281/zenodo.4906985; file 13). Bpa, 4-Benzoyl-l-phenylalanine.
(DOCX)

**S1 Table. Electrophysiological characterization of the hASIC1a TAG variant library as assessed on the SyncroPatch 384PE.** Values for $pH_{50}$, $n_H$ and Imax are shown as mean ± SD for $n \geq 3$ and as averages for $n = 1$–2. ($^{\#}$) indicates pronounced tachyphylaxis (final sweep <80% of normalized peak current). Average TE is shown in (%) as measured via GFP fluorescence on the FACS instrument. The underlying data have been deposited at zenodo.org (https://doi.org/10.5281/zenodo.4906985; files 02–06, 09, 15–28, and 33). FACS, fluorescence-activated cell sorting; hASIC1a, human acid-sensing ion channel 1a; SD, standard deviation; TE, transfection efficiency.
(DOCX)

**S2 Table. Mean current size of P2X2 and GluA2 variants evaluated in the APC screen.** Values are depicted as mean ± SD, and number in brackets indicates number of cells. The underlying data have been deposited at zenodo.org (https://doi.org/10.5281/zenodo.4906985; file 32). APC, automated patch clamp; SD, standard deviation.
(DOCX)

**S3 Table. Incorporation of ncAA photocrosslinkers in the acidic pocket of hASIC1a results in channel variants with accelerated current decay.** Displayed are mean and SD of t½; ($n$) equals number of cells. ($^{*}$) denotes significant difference between t½ of current decay compared to WT, $p < 0.05$; ($^{***}$): $p < 0.001$; Mann–Whitney test. The underlying data have been deposited at zenodo.org (https://doi.org/10.5281/zenodo.4906985; file 10). hASIC1a, human acid-sensing ion channel 1a; ncAA, noncanonical amino acid; SD, standard deviation; WT, wild type.
(DOCX)

**S4 Table. SSD curve recordings of different hASIC1a variants measured by APC.** Displayed are mean and SD of half-maximal inactivation ($pH_{50}$ SSD) and Hill slope ($n_H$) as well as number of experiments ($n$). ($^{*}$) denotes significant difference from WT, $p < 0.05$, ($^{**}$): $p < 0.005$, ($^{***}$): $p < 0.001$, ($^{****}$): $p < 0.0001$, Mann–Whitney test. The underlying data have been

deposited at zenodo.org (https://doi.org/10.5281/zenodo.4906985; files 34 and 35). APC, automated patch clamp; hASIC1a, human acid-sensing ion channel 1a; SD, standard deviation; SSD, steady-state desensitization; WT, wild type.
(DOCX)

**S5 Table. BigDyn modulation of hASIC1a WT and 6 variants containing AzF in the acidic pocket.** Cells were incubated at SSD-inducing pH for 2 minutes with or without 3 μM BigDyn before activation at pH 5.6, and the currents were normalized to the average of 2 preceding control pulses after conditioning at pH 7.6. Values are indicated as mean ± SD; (*n*) equals number of cells. (*) denotes significant difference between currents with and without 3 μM BigDyn, $p < 0.05$; (**): $p < 0.01$; (***): $p < 0.001$; ns: not significant; Mann–Whitney test. The underlying data have been deposited at zenodo.org (https://doi.org/10.5281/zenodo.4906985; file 11). AzF, 4-Azido-l-phenylalanine; hASIC1a, human acid-sensing ion channel 1a; SD, standard deviation; SSD, steady-state desensitization; WT, wild type.
(DOCX)

**S6 Table. PcTx1 modulation of hASIC1a WT and 5 variants containing AzF in the acidic pocket at different pH.** Cells were incubated with 100 nM PcTx1 for 2 minutes before activation at pH 5.6, and the current was normalized to the average of the 4 preceding and following control pulses after conditioning at pH 7.4. Values are indicated as mean ± SD; (*n*) equals number of cells. (*) denotes significant difference between currents at different pH, $p < 0.05$; (**): $p < 0.01$; ns: not significant; 1-way ANOVA with Tukey multiple comparisons test (AzF variants) or Mann–Whitney test (Bpa variants). The underlying data have been deposited at zenodo.org (https://doi.org/10.5281/zenodo.4906985; file 12). AzF, 4-Azido-l-phenylalanine; Bpa, 4-Benzoyl-l-phenylalanine; hASIC1a, human acid-sensing ion channel 1a; PcTx1, psalmotoxin 1; SD, standard deviation; WT, wild type.
(DOCX)

**S1 Text. Supporting information Methods.**
(DOCX)

## Acknowledgments

We acknowledge the FACS Core Facility at the Biotech Research & Innovation Center (University of Copenhagen) for technical support. We thank Dr. Iacopo Galleano for the synthesis of AzF- and Bpa-ME, Dr. Christian Bernsen Borg for the synthesis of big dynorphin, and members of the Pless Laboratory for comments on the manuscript. Fig 1 was created with BioRender.com.

## Author Contributions

**Conceptualization:** Nina Braun, Søren Friis, Andrea Sinz, Jacob Andersen, Stephan A. Pless.

**Data curation:** Nina Braun, Søren Friis, Stephan A. Pless.

**Formal analysis:** Nina Braun, Søren Friis, Christian Ihling.

**Funding acquisition:** Nina Braun, Andrea Sinz, Stephan A. Pless.

**Methodology:** Nina Braun, Søren Friis, Christian Ihling, Jacob Andersen.

**Project administration:** Andrea Sinz, Jacob Andersen, Stephan A. Pless.

**Resources:** Søren Friis, Christian Ihling, Andrea Sinz, Stephan A. Pless.

**Supervision:** Søren Friis, Christian Ihling, Andrea Sinz, Jacob Andersen, Stephan A. Pless.

**Validation:** Nina Braun.

**Visualization:** Nina Braun.

**Writing – original draft:** Nina Braun, Stephan A. Pless.

**Writing – review & editing:** Nina Braun, Søren Friis, Andrea Sinz, Jacob Andersen, Stephan A. Pless.

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
