## [Editor Report · Decision Letter 0]

8 Dec 2020

Dear Stephan, 

Thank you for submitting your manuscript entitled "High-throughput characterization of 309 photocrosslinker-bearing ASIC1a variants maps residues critical for channel function and pharmacology" for consideration as a Methods and Resources article by PLOS Biology.

Your manuscript has now been evaluated by the PLOS Biology editorial staff, as well as by an academic editor with relevant expertise, and I am writing to let you know that we would like to send your submission out for external peer review.

***IMPORTANT: Please note that, pending on the technical assessment delivered by the reviewers, we have decided, in agreement with the Academic Editor, that for eventual publication you would need to validate the method with at least another ion channel. This is something we will request if a revision is invited. We do not think you need to replicate the entire study with another ion channel, but just show that it will work in your system. Therefore, if you think this is a major experimental undertaking that you are not willing to do and prefer to take your submission elsewhere, do let us know, so we can terminate consideration of your submission. Do let me know if you have questions or concerns. 

In addition, before we can send your manuscript to reviewers, we need you to complete your submission by providing the metadata that is required for full assessment. To this end, please login to Editorial Manager where you will find the paper in the 'Submissions Needing Revisions' folder on your homepage. Please click 'Revise Submission' from the Action Links and complete all additional questions in the submission questionnaire.

Please re-submit your manuscript within two working days, i.e. by Dec 10 2020 11:59PM.

Kind regards,

Gabriel Gasque, Ph.D.,

Senior Editor

PLOS Biology

---

## [Decision Letter · Decision Letter 1]

29 Jan 2021

Dear Stephan,

Thank you very much for submitting your manuscript "High-throughput characterization of 309 photocrosslinker-bearing ASIC1a variants maps residues critical for channel function and pharmacology" for consideration as a Research Article at PLOS Biology. Your manuscript has been evaluated by the PLOS Biology editors, by an Academic Editor with relevant expertise, and by four independent reviewers. You will note that reviewer 2 and 3, David MacLean and Haoxing Xu, respectively, have revealed their identities. 

In light of the reviews (below), we will not be able to accept the current version of the manuscript, but we would welcome re-submission of a much-revised version that takes into account the reviewers' comments. We cannot make any decision about publication until we have seen the revised manuscript and your response to the reviewers' comments. Your revised manuscript is also likely to be sent for further evaluation by the reviewers.

We expect to receive your revised manuscript within 3 months. 

**IMPORTANT - SUBMITTING YOUR REVISION**

Your revisions should address the specific points made by the editors and by each reviewer. Having discussed these comments with the academic editor, we think you should address them in the following way:

(1) Reviewer 1 and 2’s comments should be thoroughly addressed, with additional experiments where requested.

(2) Reviewer 3 makes two major comments, one about the effects of Pctx1 on the D357AzF variant, in terms of proton sensitivity and channel desensitization, and another one about testing more substitutions at position D357. We think point 1 (the effects of Pctx1 on the D357AzF variant) is a valuable suggestion and should be addressed. Regarding the more open-ended recommendation (“it would be more informative if the authors could perform more detailed studies on the D357 position using photocrosslinkers that would introduce amino acid residues with very distinct properties.”), we do not think addressing this point experimentally is necessary for publication in our journal, assuming you will validate your method with at least another ion channel, as we requested before. However, we invite you to consider the point, as this could potentially address reviewer 4’s concern about lack of enough significant biological insights (see below). But again, this will be up to you.

(3) Reviewer 4 is critical of the work because of lack of sufficient novel biological insights and lack of transferability of the method, due to its cost. We should note that offering novel biological insights is not a condition for publication for our Methods papers. However, as mentioned above, addressing reviewer 3’s comments regarding the substitutions at position D357 could potentially address this reviewer’s concerns about lack of biological insights. Therefore, we invite you to consider this point, and if you wish to add more data, we will welcome them, but will not press for additional experiments here. Regarding the concern about cost of the equipment, we think you should add a paragraph in the Discussion to discuss transferability of the technology.

(4) I would also like to remind you that we expect you to validate the method with at least another ion channel.

(5) Finally, it is important that you update your competing interest statement to make it clear that one of the co-authors work for the APC manufacturer.

Please submit the following files along with your revised manuscript:

*Re-submission Checklist*

*Published Peer Review*

*PLOS Data Policy*

*Blot and Gel Data Policy*

Sincerely,

Gabriel Gasque, Ph.D.,

Senior Editor,

ggasque@plos.org,

PLOS Biology

REVIEWS:

Reviewer #1: The manuscript by Braun and colleagues report an extensive investigation of site-specific incorporation of non-canonical amino acids (ncaa) into the acid-sensing ion channel ASIC1a. A key point of emphasis of the paper is the development of a high throughput strategy using automated patch clamp recording. The data inform about regions of the channel protein that are involved in pH sensing and interaction with peptide ligands. The scale and scope of the study are unprecedented, and the data quality is high. I particularly like the use of 'snake plots' to summarize results. There are a few methodological and interpretational concerns that should be addressable. 

Major concerns

1. Transfection efficiency (TE) is very low for the mutants compared with WT-ASIC, but the experimental conditions for assessing this are different. According to the Methods, TE for WT-ASIC was assessed using WT-GFP, whereas TE for mutants was assessed using GFP with a nonsense mutation. Therefore, the TE value reported for mutants is a proxy for at least two phenomena (transfection efficiency and efficiency of nonsense suppression). There is value in evaluating a subset of mutants with WT-GFP (or evaluate TE for WT-ASIC with GFP-TAG) to help dissect the relative weighting of these two factors. 

2. Also related to transfection, which involved 4 separate plasmids, some additional controls may prove valuable. The main concern is that GFP fluorescence will not correlate 100% with expression of the channel. This makes it difficult to interpret the meaning of absent current in cell transfected with mutants, and therefore no conclusions can be reliable drawn based on absent current. As the authors discussed in the limitations section (page 22) a better experimental strategy would have placed both ASIC and GFP (or GFP-TAG) in the same plasmid vector. 

3. There is a great risk for sampling bias in experiments on non-sorted cells. On page 6, the authors defined specific vs unspecific incorporation based on current amplitude determined in the presence vs absence of ncaa, but the recordings on cells grown in the absence of ncaa were not done on FACS sorted cells. Although the authors addressed this on page 22, I don't think this is reliable because of the very low average TE for mutants. The authors need to consider, and probably report, the total number of recorded cells and the % of recorded cells with current vs no current for experiments investigating non-sorted populations, then describe criteria for how these data were interpreted. 

4. The criteria for excluding cells need further discussion. On page 5, a typical experiment is described in which 12 conditions across a 384-well plate are interrogated. This translates to 32 replicates per variant/condition. However, the number of replicates reported in Table S1 is 3-10 with most being less than 6. Additional information is needed to clarify the number of attempted replicates vs number of final data, and the criteria used to exclude recordings. 

Minor concerns

a. A pH titration curve should be included in Fig. 2

b. The first section of Results should have at least one paragraph break to improve readability. 

c. Throughout the text and tables, too many significant digits are presented. 

d. Page 24, 3rd line from bottom, the abbreviation 'mio.' may be a typo or a non-standard abbreviation.

Reviewer #2, David MacLean: The work by Braun et al. represents a promising direction for the field of non-canonical amino acid incorporation in ion channels, and channel biophysics in general. By combining large scale mutagenesis with FACS and automated patch clamp (APC), the authors screen just over 100 positions in human ASIC1a for incorporation of three distinct ncAAs. The approach is of interest to the many who use (or wish to use) ncAA incorporation in membrane proteins in general, and ion channels more specifically. Further, their data set on hASIC1a sites which support incorporation is gives a generous 'head start' to labs wishing to embark on this method. In addition, they use this methodology to obtain two novel scientific findings. First, that incorporation of bulky ncAAs within the acidic pocket tend to accelerate current decay and lower apparent affinity of protons (Figure 3). Second, they provide evidence that PcTx1 still binds the previously published F352L mutation but just fails to modulate it. I have one principal criticism which should be addressed and a number of minor ones, all aimed at improving this report. 

The capability of APC to probe changes in SSD (steady-state desensitization) is mentioned a few other times in the manuscript (lines 37, 92, 267, 422) yet the present data do not quite support that assertion. For example, line 265: "despite the reduced control over the conditioning pH compared to using a perfusion system with continuous flow, it was still possible to determine if BigDyn modulates hASIC1a SSD". But just a few lines above this, around line 260: "BigDyn increased rescue from pH-induced SSD in WT and all tested AzF-containing variants, although the effect was only significant for E355AzF and K356AzF (Figure 4B)". One cannot claim to have detected or evaluated an effect which is not statistically significant. I suggest the authors simply repeat the run one or two more times, to get another 3-6 data points per condition and then re-analyze the data. Also, the authors use a t-test with Welch's correction for unequal variance in this case (line 282). This is appropriate given the difference in variance (Figure 4B) but still assumes a normal distribution of samples. A better test would be a Mann Whitney U test which assumes neither normal distribution or equal variance. 

Minor

Why was GFP Y151TAG used for Azf but Y40TAG used for the others? (line 105)

The authors defined a site as having 'specific incorporation' if the peak current with ncAA > 1nA and without ncAA was < 500 pA (line 136). Unspecific incorporation was defined as currents which exceeded 1 nA in both cases. What about the case where currents were > 1 nA with ncAA but between 500 and 1000 pA without ncAA? 

Also, rooting the definition in current magnitude ignores potential changes in open probability or conductance produced by the ncAA (ie. 500 pA of current when Popen has been reduced by 80% is still a lot of channels with specific incorporation). The authors should include this caveat in their definition and consider defining it initially as apparent specific incorporation or something like that. 

Regarding the unspecific incorporation control (ie current in the absence of ncAA), the authors did not use FACS in this case because the number of GFP+ cells in the absence of ncAA is so low that downstream APC is not practical (lines 142, 489-495). Rather they just recorded a large number of transfected but unsorted cells (containing cells which were successfully transfected and cells which were not). The authors should consider in future experiments beyond this paper to transfect wild type GFP for these controls followed by FACS. At least this way they are recording from cells which are more likely to contain all plasmids while dumping those that are less likely to have been successfully transfected with all components. This might give a better estimate of unspecific incorporation. 

Figure 2E, how was relative incorporation calculated? Compared to currents in wild-type? This should be made clear. 

Lines 216-218 and Figure 3B, the authors note that certain mutants in the acidic pocket accelerate current decay. This is also reported in the abstract and a point in the discussion however no numbers are given nor statistical test performed. I appreciate the heterogeneity of the time course (line 403) but some quantification is needed to support these statements beyond the clear difference in APC traces. 

Line 287, the authors mention past reports of PcTx1 modulating the channel in a pH-dependent way. Please consider citing PMID:16284080 where the binding of radio-labelled PcTx1 is measured over a range of pHs. 

Lines 480-482, on the use of different expression plasmids, Rook et al., 2020 also used a plasmid with both tRNA and RS, as well as a C terminal GFP on the protein of interest. Please consider mentioning this and/or discussing.

Lines 497-505, the authors discuss their use of solution exchange and stacking to measure activation and SSD in ASICs. They may wish to note that ASICs are an exceptionally challenging case to apply solution switching with APC, not only because so many processes are pH-dependent but because they are so steeply pH-dependent (Hill coeffs 2-10). Most other ligand-gated channels have Hill co-effs around 1 for many of their processes. Using this HTS approach with 'better behaved' channels ought to be much easier!

Line 551, why was the ncAA used in 'acid' form for biochemistry but methyl ester for APC?

Line 590, please specify the timing of conditioning pHs used ie. how long to equilibrate before activating pH, duration of interval between stimuli etc. 

Line 612, was filtering really done at 45 Hz? Was it necessary to do this?

Line 621 versus 627, authors use Hill coefficients in the text but Hillslope in the equation. May wish to keep it consistent.

Line 637, please provide more detail on UV illumination. What was the source(s)? Intensity, distance from source, on ice, etc.

Reviewer #3, Haoxing Xu: This is an important technical advance that probed the structural-functional relationship on the acid-activated cation channel, ASIC1, using a high-throughput protocol and incorporation of non-canonical amino acids (ncAAs). Consistent with previous studies, it was found that the acid pocket domain was crucial for proton activation and channel-toxin interaction. However, the study fell in short in revealing the global trends and patterns that regulate site-specific proton activation and toxin modulation. Although three photo-crosslinkers were used, no obvious differences in terms of channel electrophysiology were found between them at most sites/positions. Hence, the molecular mechanisms underlying proton activation, channel desensitization/ tachyphylaxis, and toxin binding/modulation have remained unclear. There are some interesting observations in the study. For instance, the effects of Pctx1 on the D357AzF variant, in terms of proton sensitivity and channel desensitization, were very interesting, but the data were somehow buried in the presentation. It would be more informative if the authors could perform more detailed studies on the D357 position using photocrosslinkers that would introduce amino acid residues with very distinct properties. The three photocrosslinkers used in the study all introduce bulky groups in the sites whose actions are presumed to be charged in nature. So it was not surprising that the information revealed was somewhat limited. Overall, it is an impressive technical breakthrough, which would yield more insights into channel physiology with thoughtful experimental design. 

Reviewer #4: In this manuscript, Braun and colleagues generated 309 variants of acid-sensing ion channel 1a that contained one of three different unnatural amino acids (or non-canonical amino acids; ncAAs). Channel variants were functionally characterized via automated patch clamp (APC) and several informative mutants were further characterized by photocrosslinking via the ncAAs. Generating such a vast amount of individual variants, functionally characterizing them with high-throughput methodology and selected variants also by state-of-the-art biochemistry represents a tour-de-force of ion channel structure-function analysis. In addition, the experiments were carefully performed and results have been well documented. As such, the work clearly "demonstrates a high standard of scientific rigor in its methodology and reporting" as required for publication in PLOS Biology. My main concern rests with the importance of the study for the field.

Main comments:

1) The main aim of the study was to "establish a high-throughput protocol to conduct functional and pharmacological investigations of ncAA-containing" ion channel variants (abstract, page 2). So far, this has been limited by "a combination of low efficiency of transient transfection and limited ncAA incorporation rates" (Introduction, page 3). The principal solution to this problem was enriching by FACS the population of transiently transfected cells expressing ncAA-containing channels. Although this was a clever idea, it is not really innovative and original and does not represent a major break-through per se. 

2) Related to my first comment, in my opinion, high-throughput analysis of ion channel variants is currently not limited mainly by a lack of appropriate protocols, but rather by the fact that APC platforms are not widely used, in particular in academic environments. I suppose this is mainly due to relatively high investment and running costs. The relative scarcity of APC platforms in academia will further limit the impact of this work. 

As a side-note: since one of the authors (SF) is affiliated with the company that sells the APC platform, which was used in this study, I was wondering whether there is really no conflict of interest to declare.

3) Given that establishing a high-throughput protocol per se is not sufficient to justify publication in PLOS Biology, a journal that "features works of exceptional significance, originality, and relevance", the authors need to show that their methodology can be used to perform such works. Although they used the ASIC1a variants to analyze the interaction of ASIC1a with the opioid peptide big dynorphin and the spider toxin PcTx1, highlighting that the approach is suitable for high-throughput analysis of ion channels variants, no really new insights in the interaction of ASIC1a with big dynorphin or PcTx1 have been obtained.

One can certainly argue whether the combination of a new protocol, which is not of exceptional originality, with a very carefully executed functional and pharmacological analysis, which does not yield new insights of exceptional relevance, is still sufficient to justify publication in PLOS Biology. In my opinion, however, the lack of new insights into ASIC function and pharmacology substantially weakens the impact of this study.

Minor comments:

1) Page 6: The authors mention that they "cannot exclude the possibility of underestimating the degree of unspecific incorporation". Isn´t it actually likely that they underestimated the degree of unspecific incorporation if enriching of transfected cells grown in the absence of ncAAs by FACS was not feasible? Please comment.

2) Page 6: the authors state that they "observed mostly robust incorporation". As I understand from the next sentence, incorporation was observed at 44%-61% of the 80 positions tested. This is roughly 50%, but not "mostly".

3) At several places, the authors mention the limitation of the APC system in controlling the conditioning pH, which represents a major drawback for an ion channel like an ASIC. For a reader, who is not familiar with the APC system used, it is difficult to grasp the technical problems and potential limitations. Is it possible to explain this in more detail, perhaps in the Methods or the supplement?

4) The authors show that the F352L mutation still binds PcTx1 (Fig. 5C). But since the Western blot signal was greatly diminished, I think the authors cannot conclude that F352L "selectively abolishes the functional affects caused by PcTx1" (page 17).

5) "ASIC1a*PcTx1" and "Endogenous carboxylases" should be clearly marked by arrows in Fig. 5C.

6) On page 21, the authors speculate that the introduction of the bulkier Bpa partially occludes the acidic pocket and thus hinders binding of PcTx1. In principle, the authors could test experimentally their speculation.

---

## [Decision Letter · Decision Letter 2]

26 Apr 2021

Dear Dr Pless,

Thank you for submitting your revised Initial Research Submission entitled "High-throughput characterization of 309 photocrosslinker-bearing ASIC1a variants maps residues critical for channel function and pharmacology" for publication in PLOS Biology. I have now obtained advice from the original reviewers and have discussed their comments with the Academic Editor. 

Based on the reviews, we will probably accept this manuscript for publication, provided you satisfactorily address the data and other policy-related requests listed below my signature. 

We expect to receive your revised manuscript within two weeks. 

*Published Peer Review History*

*Early Version*

Sincerely,

Gabriel Gasque, Ph.D.,

Senior Editor,

ggasque@plos.org,

PLOS Biology

TITLE:

We would like to suggest a title that highlights more the Method and is less specific to ASIC1, since you validated the Method using another two channels:

High-throughput characterization of photocrosslinker-bearing ion channel variants to map residues critical for function and pharmacology.

ABSTRACT:

We think you should also mention in the Abstract the validation you did with other two channels.

COMPETING INTERESTS:

Please update your competing interest statement in the submission system to make it clear that one of the co-authors work for the APC manufacturer.

DATA POLICY:

Note that we do not require all raw data. Rather, we ask for all individual quantitative observations that underlie the data summarized in the figures and results of your paper. For an example see here: http://www.plosbiology.org/article/info%3Adoi%2F10.1371%2Fjournal.pbio.1001908#s5

These data can be made available in one of the following forms:

Regardless of the method selected, please ensure that you provide the individual numerical values that underlie the summary data displayed in the following figure panels: Figures 1AE, 3BC, 4BD, S1A-C, S2 (all quantitative plots, please relabel A, B, C, D, E, etc), S3 (all quantitative plots, please relabel A, B, C, D, E, etc), S4 (all quantitative plots, please relabel A, B, C, D, E, etc), S5CD, S7AB, S8C, S9B, S10B-F, S11DE, and S13.

Please also ensure that each figure legend in your manuscript includes information on where the underlying data can be found and that your supplemental data file/s has/have a legend.

We require the original, uncropped and minimally adjusted images supporting all blot and gel results reported in an article's figures or Supporting Information files. We will require these files before a manuscript can be accepted so please prepare and upload them now. Please carefully read our guidelines for how to prepare and upload this data: https://journals.plos.org/plosbiology/s/figures#loc-blot-and-gel-reporting-requirements 

BLURB:

Please provide one in our online submission system.

DATA NOT SHOWN?

Reviewer remarks:

Reviewer #1: The authors have responded fully and thoughtfully to my prior concerns. I have no remaining or new concerns. This is an elegant study.

Reviewer #2, David MacLean: The authors have satisfactorily addressed all my comments. Great work! 

Reviewer #3, Haoxing Xu: I have no further comments. 

Reviewer #4: My main concern with this manuscript was the lack of new insights into ASIC function and pharmacology. While I still have this concern, I am aware that this is a subjective evaluation and that one can come to a different conclusion. My minor concerns have all been addressed. I noticed, however, that while the conflict of interest of SF is now mentioned in the ms, it is still not mentioned in the automatically generated summary on page 1, where it reads "The authors have declared that no competing interests exist."

---

## [Editor Report · Decision Letter 3]

10 Jun 2021

Dear Stephan,

On behalf of my colleagues and the Academic Editor, Polina Lishko, I am pleased to say that we can in principle offer to publish your Methods and Resources Article "High-throughput characterization of photocrosslinker-bearing ion channel variants to map residues critical for function and pharmacology" in PLOS Biology, provided you address any remaining formatting and reporting issues. These will be detailed in an email that will follow this letter and that you will usually receive within 2-3 business days, during which time no action is required from you. Please note that we will not be able to formally accept your manuscript and schedule it for publication until you have made the required changes.

PRESS

Sincerely, 

Gabriel Gasque, Ph.D. 

Senior Editor 

PLOS Biology

ggasque@plos.org